# Adaptive Randomized Smoothing: Certified Adversarial Robustness for Multi-Step Defences

**Saiyue Lyu**[1*], **Shadab Shaikh**[1*], **Frederick Shpilevskiy**[1*], **Evan Shelhamer**[2], **Mathias Lécuyer**[1]

[1]University of British Columbia, [2]Google DeepMind

{saiyuel, shadabs3, fshpil}@cs.ubc.ca, shelhamer@deepmind.com, mathias.lecuyer@ubc.ca

[*]equal contribution

## Abstract

We propose Adaptive Randomized Smoothing (ARS) to certify the predictions of our test-time adaptive models against adversarial examples. ARS extends the analysis of randomized smoothing using $f$-Differential Privacy to certify the adaptive composition of multiple steps. For the first time, our theory covers the sound adaptive composition of general and high-dimensional functions of noisy inputs. We instantiate ARS on deep image classification to certify predictions against adversarial examples of bounded $L_\infty$ norm. In the $L_\infty$ threat model, ARS enables flexible adaptation through high-dimensional input-dependent masking. We design adaptivity benchmarks, based on CIFAR-10 and CelebA, and show that ARS improves standard test accuracy by 1 to 15% points. On ImageNet, ARS improves certified test accuracy by up to 1.6% points over standard RS without adaptivity. Our code is available at https://github.com/ubc-systopia/adaptive-randomized-smoothing.

## 1 Introduction

Despite impressive accuracy, deep learning models still show a worrying susceptibility to adversarial attacks. Such attacks have been shown for a large number of tasks and models (Costa et al., 2023; Chakraborty et al., 2018), including areas where security and safety are critical such as fraud detection (Pumsirirat and Liu, 2018) or autonomous driving (Cao et al., 2021).

Several rigorous defences have been proposed to certify robustness. Randomized Smoothing (RS) (Lécuyer et al., 2019; Cohen et al., 2019) does so by averaging predictions over noisy versions of the input at test time, and as such can scale to large deep learning models. However, RS has its limitations: it is inflexible and either degrades accuracy or only certifies against small attacks.

To address these shortcomings and improve robustness, there has been a recent push to develop defences that adapt to inputs at test time Croce et al. (2022), including for RS (Alfarra et al., 2022a; Súkeník et al., 2021; Hong et al., 2022). Most such adaptive defences are heuristic, unproven, and subject to improved attacks (Croce et al., 2022; Alfarra et al., 2022a; Hong et al., 2022), running the risk of reverting to a hopeless cat and mouse game with attackers (Athalye et al., 2018; Tramer et al., 2020), or only provide limited adaptivity (Súkeník et al., 2021; Hong et al., 2022) and gain (§5).

We (re)connect RS to Differential Privacy (DP), after its abandonment for a tighter analysis via hypothesis testing (Cohen et al., 2019), and introduce **Adaptive Randomized Smoothing (ARS)** to provide test-time adaptivity while preserving rigorous bounds. Specifically, we analyze RS through the lens of $f$-Differential Privacy ($f$-DP), and use this connection to leverage a key strength of DP: the end-to-end analysis of multi-step adaptive computation using composition results (§2).

We use ARS to design two-step defence against $L_\infty$ adversaries on image classification (Figure 1), which is a challenging setting for RS Blum et al. (2020). The first step computes an input mask that

38th Conference on Neural Information Processing Systems (NeurIPS 2024).

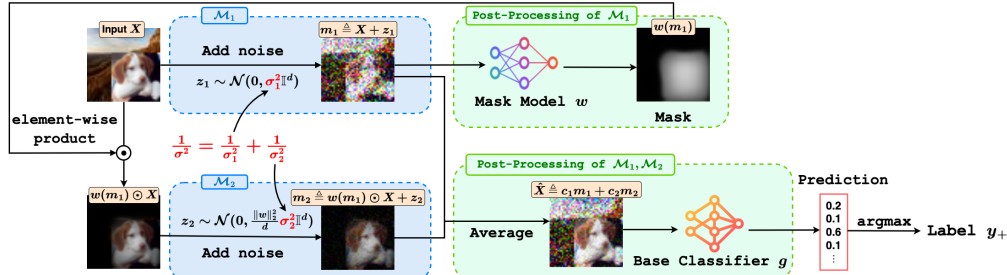

Figure 1: Two-step ARS for $L_\infty$-bounded attacks. Step $\mathcal{M}_1$ adds noise to input $X$ and post-processes the result into a mask $w(m_1)$. Step $\mathcal{M}_2$ takes masked input $w(m_1) \odot X$ and adds noise to get $m_2$. Base classifier $g$ post-processes a weighted average of $m_1, m_2$ to output a label. RS reduces to $\sigma_2 = \sigma$ and $w(.) = 1$ (no $\mathcal{M}_1$).

focuses on task-relevant information. This reduces the dimension of the input, which is then passed to the second step for prediction. Thanks to this adaptive dimension reduction, the second step makes its prediction on a less noisy image, improving the performance and certification radius (§3).

We evaluate our adaptive randomized smoothing method in three settings (§4). For image classification, we first design a challenging benchmark based on CIFAR-10, and we show that ARS can improve accuracy by up to 15%. For spatially-localized face attribute classification on the CelebA dataset, we show that ARS improves accuracy by up to 7.7%. For large-scale image classification on ImageNet, ARS maintains accuracy and improves certified accuracy by up to 1.6%.

## 2 Theory for Adaptivity via Differential Privacy

After introducing the necessary background and known results on RS and DP (§2.1), we reconnect RS to its DP roots by showing that the tight analysis of Cohen et al. (2019) can be seen as PixelDP (Lécuyer et al., 2019) using $f$-Differential Privacy (Dong et al., 2019; 2022), a hypothesis testing formulation of DP (§2.2). This connection lets us leverage composition results for $f$-DP to analyze multi-step approaches for provable robustness to adversarial examples, which we name Adaptive Randomized Smoothing (ARS) (§2.3). We leverage ARS to design and analyze a two-step defence against $L_\infty$-bounded adversaries (§2.4), which we then instantiate as a deep network (§3).

### 2.1 Related Work: Adversarial Robustness, Randomized Smoothing, and Differential Privacy

**Adversarial Examples** (Szegedy et al., 2014): Consider a classifier $g : \mathcal{X} \to \mathcal{Y}$, and input $X$. An adversarial example of radius $r$ in the $L_p$ threat model, for model $g$ on input $X$, is an input $X + e$ such that $g(X + e) \neq g(X)$, where $e \in B_p(r)$, where $B_p(r) \triangleq \{x \in \mathbb{R}^d : \|x\|_p \leq r\}$ is the $L_p$ ball of radius $r$. These inputs or attacks are made against classifiers at test time. For more on the active topics of attack and defence, we refer to surveys (Li et al., 2023; Costa et al., 2023; Chakraborty et al., 2018) on current attacks and provable defences. In general, provable defences do not focus on the largest-scale highest-accuracy classifiers, with the notable exception of randomized smoothing.

**Randomized Smoothing (RS)** (Lécuyer et al., 2019; Cohen et al., 2019) is a scalable approach to certifying model predictions against $L_2$-norm adversaries. Specifically, it certifies robustness to *any* attack $\in B_2(r_X)$. The algorithm randomizes a base model $g$ by adding spherical Gaussian noise to its input, and produces a smoothed classifier that returns the class with highest expectation over the noise: $y_+ \triangleq \arg\max_{y \in \mathcal{Y}} \mathbb{P}_{z \sim \mathcal{N}(0, \sigma^2 \mathbb{I}^d)}(g(X + z) = y)$. The tightest analysis from Cohen et al. (2019) uses hypothesis testing theory to show that, with $\underline{p_+}, \overline{p_-} \in [0, 1]$ such that $\mathbb{P}(g(X + z) = y_+) \geq \underline{p_+} \geq \overline{p_-} \geq \max_{y_- \neq y_+} \mathbb{P}(g(X + z) = y_-)$, the certificate size $r_X$ for prediction $y_+$ is:

$$r_X = \frac{\sigma}{2}\big(\Phi^{-1}(\underline{p_+}) - \Phi^{-1}(\overline{p_-})\big), \tag{1}$$

where $\Phi^{-1}$ is the inverse standard Gaussian CDF, $\underline{p_+}$ lower-bounds the probability of $g(X + z) = y_+$ (the most probable class), and $\overline{p_-}$ upper-bounds the probability of other classes.

While sound, RS is static during testing even though attacks may adapt. Recent work aims to make RS adapt at test time (Súkeník et al., 2021; Alfarra et al., 2022a; Hong et al., 2022). While pioneering, these works are restricted in either their soundness or their degree of adaptation and resulting improvement. Súkeník et al. (2021) soundly adapt the variance for RS by the distance between test and train inputs. However, this only provides minimal adaptivity, with only minor improvement to certification. Alfarra et al. (2022a) adapt the variance for RS to each test input, but the analysis is not end-to-end, and hence not sound (Súkeník et al., 2021) (except for their memory-based edition, but this requires storage of all examples). UniCR (Hong et al., 2022) adapts the noise distribution for RS, primarily to the data distribution during training, and optionally to input during testing. The train-time adaptation is sound, but the test-time adaptation is not due to the same issue raised by Súkeník et al. (2021). We propose ARS to advance certified test-time adaptivity: our approach is sound and high-dimensional to flexibly adapt the computation of later steps conditioned on earlier steps by differential privacy theory.

**Differential Privacy (DP)** is a rigorous notion of privacy. A randomized mechanism $\mathcal{M}$ is $(\epsilon, \delta)$-DP if, for any neighbouring inputs $X$ and $X'$, and any subset of possible outputs $\mathcal{Y} \subset \text{Range}(\mathcal{M})$, $\mathbb{P}(\mathcal{M}(X) \in \mathcal{Y}) \leq e^\epsilon \mathbb{P}(\mathcal{M}(X') \in \mathcal{Y}) + \delta$. Following Lécuyer et al. (2019), we define neighbouring based on $L_p$ norms: $X$ and $X'$ in $\mathbb{R}^d$ are $L_p$ neighbours at radius $r$ if $X - X' \in B_p(r)$.

RS was initially analyzed using $(\epsilon, \delta)$-Differential Privacy (Lécuyer et al., 2019). Intuitively, the randomized classifier $\mathcal{M}(X) \triangleq g(X + z)$, $z \sim \mathcal{N}(0, \sigma^2 \mathbb{I}^d)$ acts as a privacy preserving mechanism (the Gaussian mechanism) that provably "hides" small variations in $X$. This privacy guarantee yields a robustness certificate for the expected predictions.

**$f$-DP** (Dong et al., 2019) is a notion of privacy that extends $(\epsilon, \delta)$-DP, and defines privacy as a bound on the power of hypothesis tests. Appendix A provides more details on $f$-DP. The main result we leverage is Theorem 2.7 of Dong et al. (2022) that for a Gaussian mechanism $\mathcal{M}(X) = \theta(X) + z$, $z \sim \mathcal{N}(0, \frac{r^2}{\mu^2})$, such that for any neighbouring $X, X'$, $\theta(X) - \theta(X') \in B_2(r)$ (i.e., the $L_2$ sensitivity of $\theta$ is $r$), we have that $\mathcal{M}$ is $G_\mu$-DP with the function $G_\mu$ defined as:

$$G_\mu(\alpha) = \Phi\Big(\Phi^{-1}(1 - \alpha) - \mu\Big), \text{ for } \alpha \in [0, 1]. \tag{2}$$

We leverage two key properties of $f$-DP. First, $f$-DP is resilient to post-processing. That is, if mechanism $\mathcal{M}$ is $f$-DP, proc $\circ \mathcal{M}$ is also $f$-DP. Second, $f$-DP is closed under adaptive composition. We refer to §3 in Dong et al. (2022) for the precise definition and use their Corollary 3.3: the adaptive composition of two Gaussian mechanisms $G_{\mu_1}$-DP and $G_{\mu_2}$-DP is itself $G_\mu$-DP with

$$\mu = \sqrt{\mu_1^2 + \mu_2^2}. \tag{3}$$

$f$-DP is distinct from the $f$-divergence from information theory. Dvijotham et al. (2020) use $f$-divergence bounds between the noise distribution centred on the original input and centred on any perturbed input. This improves RS by broadening the noise distributions and norm-bounds on the adversary that RS can support. In contrast we focus on $f$-DP, which captures enough information to reconstruct any divergence by post-processing (Proposition B.1. in Dong et al. (2019)). Our main objective is different: we leverage $f$-DP *composition* properties to enable multi-step deep learning architectures that adapt to the input at test time with robustness guarantees.

## 2.2 Randomized Smoothing from $f$-DP

We reconnect RS with DP, using $f$-DP to yield results as strong as that of Equation (1). We start with a general robustness result on $f$-DP classifiers, which we later build on for our main result.

**Proposition 2.1** ($f$-DP Robustness). *Let $\mathcal{M} : \mathcal{X} \to \mathcal{Y}$ be $f$-DP for $B_p(r)$ neighbourhoods, and let $M_S : X \to \arg\max_{y \in \mathcal{Y}} \mathbb{P}(\mathcal{M}(X) = y)$ be the associated smoothed classifier. Let $y_+ \triangleq M_S(X)$ be the prediction on input $X$, and let $\underline{p_+}, \overline{p_-} \in [0, 1]$ be such that $\mathbb{P}(\mathcal{M}(X) = y_+) \geq \underline{p_+} \geq \overline{p_-} \geq \max_{y_- \neq y_+} \mathbb{P}(\mathcal{M}(X) = y_-)$. Then:*

$$f(1 - \underline{p_+}) \geq 1 - f(\overline{p_-}) \Rightarrow \forall e \in B_p(r), \ M_S(X + e) = y_+$$

*Proof.* See Appendix B1. $\square$

Let us now instantiate Proposition 2.1 for Gaussian RS (see §2.1):

**Corollary 2.2** (RS from $f$-DP). *Let $\mathcal{M} : X \to g(X + z)$, $z \sim \mathcal{N}(0, \sigma^2 \mathbb{I}^d)$, and $M_S : X \to \arg\max_{y \in \mathcal{Y}} \mathbb{P}(\mathcal{M}(X) = y)$ be the associated smooth model. Let $y_+ \triangleq M_S(X)$ be the prediction on input $X$, and let $\underline{p_+}, \overline{p_-} \in [0, 1]$ be such that $\mathbb{P}(\mathcal{M}(X) = y_+) \geq \underline{p_+} \geq \overline{p_-} \geq \max_{y_- \neq y_+} \mathbb{P}(\mathcal{M}(X) = y_-)$. Then $\forall e \in B_2(r_x)$, $M_S(X + e) = y_+$, with:*

$$r_X = \frac{\sigma}{2}\big(\Phi^{-1}(\underline{p_+}) - \Phi^{-1}(\overline{p_-})\big).$$

*Proof.* See Appendix B2. *Sketch:* $\mathcal{M}$ is a Gaussian mechanism, and is $G_{\frac{r}{\sigma}}$-DP for any $r$ ($B_2(r)$ neighbourhood). We apply Proposition 2.1 and maximize $r$ such that $G_{\frac{r}{\sigma}}(1 - \underline{p_+}) \geq 1 - G_{\frac{r}{\sigma}}(\overline{p_-})$. $\quad\square$

## 2.3 Adaptive Randomized Smoothing

While Proposition 2.1 is new, so far we have only used it to reprove the known result of Corollary 2.2. So why is this connection between $f$-DP and robustness useful? Our key insight is that we can leverage adaptive composition results at the core of DP algorithms to *certify multi-step methods that adapt to their inputs at test time*. Such adaptive defences have seen recent empirical interest, but either lack formal guarantees, or provide only limited adaptivity in practice (§5). For the first time we derive a *sound and high-dimensional* adaptive method for certification.

We formalize adaptive multi-step certification as follows. Consider $k$ randomized Gaussian mechanisms $\mathcal{M}_1, \ldots, \mathcal{M}_k$ (our adaptive steps), such that $m_i \sim \mathcal{M}_i(X | m_{<i})$, and for all $r \geq 0$ we have that $\mathcal{M}_i$ is $G_{r/\sigma_i}$-DP for the $B_2(r)$ neighbouring definition. Note that the computation $\mathcal{M}_i$ can depend on previous results, as long as it is $G_{r/\sigma_i}$-DP. Further consider a (potentially randomized) post-processing classifier $g(m_1, \ldots, m_k) = y \in \mathcal{Y}$.

**Theorem 2.3** (Main result: Adaptive RS). *Using definitions above, let $\mathcal{M} : X \to g(m_1, \ldots, m_k) \in \mathcal{Y}$, $(m_1, \ldots, m_k) \sim (\mathcal{M}_1(X), \ldots, \mathcal{M}_k(X | m_{<k}))$, and the associated smoothed model be $M_S : X \to \arg\max_{m \in \mathcal{Y}} \mathbb{P}(\mathcal{M}(X) = y)$. Let $y_+ \triangleq M_S(X)$ be the prediction on input $X$, and let $\underline{p_+}, \overline{p_-} \in [0, 1]$ be such that $\mathbb{P}(\mathcal{M}(X) = y_+) \geq \underline{p_+} \geq \overline{p_-} \geq \max_{y_- \neq y_+} \mathbb{P}(\mathcal{M}(X) = y_-)$. Then $\forall e \in B_2(r_x)$, $M_S(X + e) = y_+$, with:*

$$r_X = \frac{1}{2\sqrt{\sum_{i=1}^{k} \frac{1}{\sigma_i^2}}}\Big(\Phi^{-1}(\underline{p_+}) - \Phi^{-1}(\overline{p_-})\Big).$$

*Proof.* By adaptive composition of Gaussian DP mechanisms (Equation (3)), $\mathcal{M}$ is $G_\mu$-DP with $\mu = \sqrt{\sum_{i=1}^{k} \frac{r^2}{\sigma_i^2}} = r\sqrt{\sum_{i=1}^{k} \frac{1}{\sigma_i^2}}$. We can then apply Corollary 2.2 with $\sigma = 1/\sqrt{\sum_{i=1}^{k} \frac{1}{\sigma_i^2}}$. $\quad\square$

We focus on Gaussian RS, but a similar argument applies to general $f$-DP mechanisms for which we can compute $f_i$ at any $r$, and the composition $f_i \otimes \cdots \otimes f_k$, potentially using numerical techniques such as that of Gopi et al. (2021). For Gaussian noise, Theorem 2.3 leverages strong results from DP to provide a perhaps surprising result: there is no cost to adaptivity, in the sense that $k$ independent measurements of input $X$ with Gaussian noise (without adaptivity) of respective variance $\sigma_i^2$ can be averaged to one measurement of variance $\sigma^2 = 1/\sum_{i=1}^{k} \sigma_i^{-2}$. To show this, we can use a weighted average to minimize variance (see e.g., Equation 4 in Honaker (2015)), with $c_j = \sigma_j^{-2}/\sum_{i=1}^{k} \sigma_i^{-2}$ yielding $\sigma^2 = \sum_{j=1}^{k} c_j^2 \sigma_j^2 = \sum_{j=1}^{k} \sigma_j^{-2}/\big(\sum_{i=1}^{k} \sigma_i^{-2}\big)^2 = 1/\sum_{i=1}^{k} \sigma_i^{-2}$. The ARS $r_X$ from Theorem 2.3 is identical to that of one step RS from Corollary 2.2 using this variance: *adaptivity over multi-step computation comes with no decrease in certified radius.*

## 2.4 ARS against $L_\infty$-Bounded Adversaries

How can we leverage the multi-step adaptivity from Theorem 2.3 to increase certified accuracy? We focus on two-step certified defence against $L_\infty$-bounded attacks to increase accuracy by adaptivity. Previous work already notes that RS applies to $L_\infty$-bounded attackers (Lécuyer et al., 2019; Cohen

et al., 2019), using the fact that $\forall X \in \mathbb{R}^d, \|X\|_2 \leq \sqrt{d}\|X\|_\infty$, and hence that $X - X' \in B_\infty(r^\infty) \Rightarrow X - X' \in B_2(\sqrt{d} \cdot r^\infty)$. Using Corollary 2.2, this yields:

$$r_X^\infty = \frac{\sigma}{2\sqrt{d}}\big(\Phi^{-1}(\underline{p_+}) - \Phi^{-1}(\overline{p_-})\big). \tag{4}$$

While $L_\infty$-specific RS theory exists (Yang et al., 2020), further work by Blum et al. (2020) has found that Gaussian RS performs advantageously in practice. However, Blum et al. (2020); Kumar et al. (2020); Wu et al. (2021) show that the $\sqrt{d}$ dependency cannot be avoided for a large family of distributions, leading the authors to speculate that RS might be inherently limited for $L_\infty$ certification of predictions on high dimensional images. To side-step this issue, we use two-steps adaptivity to first select subsets of the image important to the classification task (thereby reducing dimension), and then make the prediction based on the selected subset. Formally:

**Proposition 2.4** (Adaptive RS for $L_\infty$). *Define the following pair of (adaptive) mechanisms:*

$$\mathcal{M}_1 : X \to X + z_1 \triangleq m_1, \qquad z_1 \sim \mathcal{N}(0, \sigma_1^2 \mathbb{I}^d) \tag{5}$$

*Then, with any function $w : \mathbb{R}^d \to [0,1]^d$ (interpreted as a mask):*

$$\mathcal{M}_2 : (X, m_1) \to w(m_1) \odot X + z_2 \triangleq m_2, \qquad z_2 \sim \mathcal{N}(0, \frac{\|w(m_1)\|_2^2}{d}\sigma_2^2 \mathbb{I}^d) \tag{6}$$

*where $\odot$ is the element-wise product; and the final prediction function $g : m_1, m_2 \to \mathcal{Y}$.*

*Consider the mechanism $\mathcal{M}$ that samples $m_1 \sim \mathcal{M}_1$, then $m_2 \sim \mathcal{M}_2$, and finally outputs $g(m_1, m_2)$; and the associated smoothed classifier $M_S : X \to \arg\max_{y \in \mathcal{Y}} \mathbb{P}(\mathcal{M}(X) = y)$. Let $y_+ \triangleq M_S(X)$ be the prediction on input $X$, and let $p_+, \overline{p_-} \in [0,1]$ be such that $\mathbb{P}(\mathcal{M}(X) = y_+) \geq \underline{p_+} \geq \overline{p_-} \geq \max_{y_- \neq y_+} \mathbb{P}(\mathcal{M}(X) = y_-)$. Then $\forall e \in B_\infty(r_X^\infty), M_S(X + e) = y_+$, with:*

$$r_X^\infty = \frac{1}{2\sqrt{d\big(\frac{1}{\sigma_1^2} + \frac{1}{\sigma_2^2}\big)}}\big(\Phi^{-1}(\underline{p_+}) - \Phi^{-1}(\overline{p_-})\big). \tag{7}$$

*Proof.* Consider any $X, X'$ s.t. $X - X' \in B_\infty(r^\infty)$. We analyze $\mathcal{M}_1$ and $\mathcal{M}_2$ in turn. $\|X - X'\|_2 \leq \sqrt{d}\|X - X'\|_\infty$, so $X - X' \in B_2(\sqrt{d}r^\infty)$, and $\mathcal{M}_1$ is $G_{\mu_1}$-DP with $\mu_1 = \frac{r^\infty \sqrt{d}}{\sigma_1}$.

$\|w(m_1) \odot X - w(m_1) \odot X'\|_2 = \|w(y_1) \odot (X - X')\|_2 \leq \|w(y_1)\|_2\|X - X'\|_\infty$ so $X - X' \in B_2(\|w(y_1)\|_2 r^\infty)$ and $\mathcal{M}_2$ is $G_{\mu_2}$-DP with $\mu_2 = \frac{\|w(y_1)\|_2 r^\infty}{\|w(y_1)\|_2 \sigma_2/\sqrt{d}} = \frac{r^\infty \sqrt{d}}{\sigma_2}$.

Applying Theorem 2.3 with $\sqrt{\frac{(r^\infty)^2 d}{\sigma_1^2} + \frac{(r^\infty)^2 d}{\sigma_2^2}} = r^\infty \sqrt{d\big(\frac{1}{\sigma_1^2} + \frac{1}{\sigma_2^2}\big)}$ concludes the proof. $\square$

**Important remarks. 1.** $w(.)$ is a masking function, adaptively reducing (if $w_i(m_1) \ll 1$) the value of $X_i$ and thereby the attack surface of an $L_\infty$ attacker. This reduces the effective dimension of the input to $\mathcal{M}_2$. **2.** Reducing the dimension enables a reduction in the noise variance in $\mathcal{M}_2$, at fixed privacy guarantee $G_{\mu_2}$. The variance reduction is enabled *for all dimensions in the input*, even those that are not masked ($w_i(m_1) \approx 1$). As a result, the variance of the noise in $\mathcal{M}_2$ scales as $\|w(m_1)\|_2^2 \leq d$. The more masking, the lower the variance. It may help to consider the change of variables $\sigma \leftarrow \sigma/\sqrt{d}$ in Equation (4), and $\sigma_{1,2} \leftarrow \sigma_{1,2}/\sqrt{d}$ in Proposition 2.4, to remove $d$ from $r_X^\infty$ and scale the noise variance with $d$. For RS (Equation (4)), the noise variance scales as $d$. For ARS, only Equation (5) (the first step) suffers from variance scaled by $d$, while the second step's variance (Equation (6)) scales as $\|w(m_1)\|_2^2$, which can be much smaller than $d$ when a large part of the image is masked. Reduced variance translates into higher accuracy, as well as $\underline{p_+}$ and $\overline{p_-}$ being further apart, for a larger $r_X^\infty$. **3.** The variance reduction due to masking applies in the translation from the $B_\infty(r^\infty)$ bound on the attack to the $B_2(r)$ sensitivity used in ARS. This variance reduction would not apply to an $L_2$-bounded adversary (an attack that only changes pixels with mask values of one yields no sensitivity improvement). Hence, our two-steps ARS architecture for $L_\infty$-bounded adversaries does not reduce to bounding $L_\infty$ with $L_2$ as the traditional RS application does, and our gains come explicitly from variance reduction enabled by adaptive masking against an $L_\infty$ attack.

# 3 Two-Step ARS for $L_\infty$ Certification of Image Classification

Figure 1 shows our deep learning architecture based on Proposition 2.4. The first step, $\mathcal{M}_1$, adds noise to input $X$ and post-processes the result into a mask $w(m_1)$. The second step, $\mathcal{M}_2$, takes masked input $w(m_1) \odot X$ and adds noise. Finally, the base classifier $g$ post-processes a weighted average of $m_1, m_2$ to output a label. The whole model is trained end-to-end on the classification task. In RS, only the path going through $\mathcal{M}_2$ is present. This is equivalent to setting $\sigma_2 = \sigma$ and $w(.) = 1$, with no $\mathcal{M}_1$. In both cases, the final predictions are averaged over the noise to create the smoothed classifier. The ARS architecture introduces several new components, which we next describe.

**Budget Splitting:** the noise budget $\sigma$ (Figure 1; red) is split to assign noise levels to the two steps $\mathcal{M}_1$ and $\mathcal{M}_2$. We parameterize ARS with the same $\sigma$ as standard RS then split it by the $f$-DP composition formula from Equation (3). In practice, we assign $\sigma_1 \geq \sigma$ to $\mathcal{M}_1$, and then $\sigma_2 = 1/\sqrt{\frac{1}{\sigma^2} - \frac{1}{\sigma_1^2}}$. We set $\sigma_1$ by either fixing it to a constant or learning it end-to-end.

**Masking:** the mask model $w(\cdot)$ takes the noisy image from $\mathcal{M}_1$ and predicts a weighting (one value in $[0, 1]$ per input pixel) that is multiplied with the input element-wise (denoted $\odot$ in Proposition 2.4). The model is a U-Net architecture, which makes pixel-wise predictions, and acts as a post-processing of $\mathcal{M}_1$ in the $f$-DP analysis. Our masking enables test-time adaptivity to reduce the noise variance for $\mathcal{M}_2$, via the mask's dependence on the input through $m_1$.

**Mechanism output averaging:** to fully leverage both steps' information, we take a weighted average of the outputs $m_1$ and $m_2$ before passing the result to the base classifier $g$. For a particular input pixel $i$, denote $X_i$ the value of pixel, $w_i \in [0, 1]$ its mask weight (we omit the explicit dependency on $m_1$ in $w$ for compactness), and $m_{1,i}, m_{2,i}$ the respective values output by $\mathcal{M}_1$ and $\mathcal{M}_2$. Then, the final value of pixel $i$ in the averaged input will be $\hat{X}_i \triangleq c_{1,i} m_{1,i} + c_{2,i} m_{2,i}$.

We set $c_{1,i}, c_{2,i}$ such that $\hat{X}_i$ is the unbiased estimate of $X_i$ with smallest variance. First, we set $c_{1,i} + w_i c_{2,i} = 1$, such that $\mathbb{E}[\hat{X}_i] = c_{1,i} X_i + c_{2,i} w_i X_i = X_i$. Second, we minimize the variance. Notice that $\mathbb{V}[\hat{X}_i] = c_{1,i}^2 \sigma_1^2 + c_{2,i}^2 \|w\|_2^2 \sigma_2^2 = (1 - w_i c_{2,i})^2 \sigma_1^2 + c_{2,i}^2 \|w\|_2^2 \sigma_2^2$: this is a convex function in $c_{2,i}$ minimized when its gradient in $c_{2,i}$ is zero. Plugging back into the constraint to get $c_{1,i}$, we obtain the following weights: $c_{1,i} = \frac{\|w\|_2^2 \sigma_2^2}{\sigma_1^2 w_i^2 + \|w\|_2^2 \sigma_2^2}$, and $c_{2,i} = \frac{\sigma_1^2 w_i}{\sigma_1^2 w_i^2 + \|w\|_2^2 \sigma_2^2}$.

The averaged noisy input $\hat{X}$ is finally fed to the base classifier $g$ for prediction. The smoothed classifier $M_S$ averages predictions (over noise draws) over the entire pipeline. The parameters of $w$ and $g$ (and $\sigma_1$ if not fixed) are learned during training and are fixed at inference/certification time.

# 4 Experiments

We evaluate on standard and $L_\infty$-certified test accuracy. Certified accuracy at radius $r^\infty$ is the percentage of test samples that are correctly classified **and** have an $L_\infty$ certificate radius $r_X^\infty \geq r^\infty$. Standard accuracy is obtained for $r^\infty = 0$.

**Datasets** We evaluate on CIFAR-10 (Krizhevsky, 2009) in §4.1, CelebA (Liu et al., 2015) (specifically the unaligned HD-CelebA-Cropper edition) in §4.2, and ImageNet (Deng et al., 2009) in §4.3. We measure adaptivity on CIFAR-10 and CelebA by designing challenging benchmarks requiring adaptivity, and measure scalability on ImageNet.

**Models** We choose the standard ResNet (He et al., 2016) models as base classifiers $g$ with ResNet-110 for CIFAR-10 and ResNet-50 for CelebA and ImageNet. For ARS, our mask model $w$ is a simplified U-Net (Ronneberger et al., 2015) (see Appendix C.1 for details). For the noise budget, we find that a fixed budget split performs reliably, and so in all experiments we split by $\sigma_1 = \sigma_2 = \sqrt{2}\sigma$.

**Methods** We compare to standard and strong static methods, design a baseline specifically for our masking approach, and evaluate the only sound input-dependent methods prior to ARS. *Cohen et al.* is the standard approach to RS (Cohen et al., 2019). *UniCR* (Hong et al., 2022) learns the noise distribution for RS during training but is static during testing (while they propose an input-adaptive variant, it is not sound so we restrict our comparison to the training variant). We tune hyper-parameters, and perform a grid search for $\beta$ (the parameter of the noise distribution) to maximize certified accuracy. We find that $\beta = 2$ (Gaussian), or $\beta = 2.25$ (close to a Gaussian,

| Setting/Approach | Cohen et al. | Static Mask | UniCR | Súkeník et al.$^\triangle$ | ARS $^\triangle$ |
|---|---|---|---|---|---|
| $\sigma = 0.25,\ k = 32$ | 70.6 (1.1) | **73.9 (0.8)** | 69.8 (1.4) | 68.6 (2.8) | 72.6 (0.9) |
| $\sigma = 0.5,\ \ k = 32$ | 63.6 (2.0) | **64.8(0.9)** | 62.8 (1.4) | 59.1 (1.6) | 64 (1.4) |
| $\sigma = 1.0,\ \ k = 32$ | 48 (0.7) | 47.3 (1.3) | 46.1 (0.9) | 44.6 (1.0) | **49.3 (0.6)** |
| $\sigma = 0.25,\ k = 48$ | 71.6 (1.0) | 72 (2.0) | 69.7 (0.8) | 65 (0.7) | **75.5 (1.0)** |
| $\sigma = 0.5,\ \ k = 48$ | 64.3 (0.2) | 64.1 (1.6) | 60.3 (0.6) | 53.5 (1.8) | **66 (1.6)** |
| $\sigma = 1.0,\ \ k = 48$ | 42.5 (2.1) | 45.1 (1.2) | 44.3 (0.2) | 34.1 (1.0) | **47.6 (2.0)** |
| $\sigma = 0.25,\ k = 64$ | 71.6 (0.9) | 73.1 (3.2) | 67.8 (0.5) | 64.1 (0.8) | **77 (1.8)** |
| $\sigma = 0.5,\ \ k = 64$ | 63 (1.6) | 62.5 (1.7) | 58.7 (0.2) | 45.1 (1.1) | **69.9 (1.2)** |
| $\sigma = 1.0,\ \ k = 64$ | 41.3 (1.8) | 40.0 (0.5) | 42.2 (0.6) | 26.5 (0.7) | **50.4 (2.5)** |
| $\sigma = 0.25,\ k = 96$ | 65.3 (1.6) | 71.8 (1.3) | 68.8 (1.8) | 45.5 (0.9) | **78.3 (2.2)** |
| $\sigma = 0.5,\ \ k = 96$ | 56.6 (2.4) | 59.5 (1.4) | 59.7 (1.3) | 10.8 (2.3) | **69.8 (1.2)** |
| $\sigma = 1.0,\ \ k = 96$ | 33.8 (3.8) | 36.9 (0.5) | 41.3 (2.4) | 10.4 (0.3) | **56.3 (2.3)** |

Table 1: **Standard Accuracy ($r = 0$) on CIFAR-10 (20kBG).** Our 20kBG benchmark places CIFAR-10 images on larger background images. We report the mean accuracy and standard deviation over three seeds. ARS achieves higher accuracy across noise $\sigma$ and input dimension $k$ ($^\triangle$ indicates adaptivity). We provide results with more $\sigma$ levels in Appendix D.

with smaller tails), to perform best in high and low $(k, \sigma)$ values, respectively (see Appendix D) for details. *Static Mask* is our baseline that learns a fixed mask during training that does not adapt during testing. The mask is directly parameterized as pixel-wise weights that we multiply with the input and optimize jointly with the base classifier. Relative to ARS in Figure 1, this removes $\mathcal{M}_1$, sets $\sigma_2 = \sigma$, and makes $w(.) = W$ static parameters rather than an adaptive prediction. *Súkeník et al. (2021)* conditions the variance $\sigma$ for RS on the input and is therefore test-time adaptive. We use code provided by the authors as is. Comparing ARS to the static baselines measures the value of test-time adaptivity, and comparing ARS to the variance adaptivity of Súkeník et al. measures the importance of more high-dimensional and expressive adaptation.

## 4.1 CIFAR-10 Benchmark: Classification with Distractor Backgrounds

Input dimension is a key challenge in $L_\infty$ certification using RS (see §2.4). We design our 20kBG benchmark to vary this parameter without affecting the task: we superimpose CIFAR-10 images onto a larger distractor background from the $20k$ background images dataset Li et al. (2022a). The backgrounds are split into train and test sets, and resized to $k \times k$ pixels (where $k \geq 32$). The CIFAR-10 image (of dimensions $32 \times 32 \times 3$) is then placed at random along the edges of the background image to maximize spatial variation. The spurious background increases the dimension ($= k \times k \times 3$) of the input when $k > 32$, making $L_\infty$ certification with RS more challenging, but is uninformative for the task of CIFAR-10 by construction. Our mask model ($\mathcal{M}_1$) needs to learn to ignore the background to reduce the effective dimension of the input. For computational reasons, we run all certification results on a 200 sample subset of the CIFAR-10 test set. Appendix D shows extended results, details about hyperparameter tuning (C.3), and results on larger test-sets (D.1). We set the failure probability of the certification procedure to 0.05, use 100 samples to select the most probable class, and 50,000 samples for the Monte Carlo estimate of $\underline{p_+}$.

Table 1 summarizes the standard accuracy ($r = 0$) of each approach on the different settings. We vary $\sigma$ in $\{0.25, 0.5, 1.0\}$ and $k$ in $\{32, 48, 64, 96\}$ to show the effect of noise levels and dimensionality on the accuracy of different approaches ($k = 32$ corresponds to original CIFAR-10 images). Figure 2 shows the certified test accuracy at different radii $r^\infty$ for ARS and all baselines we consider.

We make three observations. First, ARS outperforms all the baselines under distractor backgrounds ($k > 32$). Static mask is slightly better at $k = 32$, probably because CIFAR-10 images lack enough "irrelevant" information for ARS to discard. This is precisely why we introduce this benchmark, where we explicitly add such irrelevant information to input images. Indeed, at $k > 32$, we observe that the standard accuracy of ARS improves, translating to an improved certified accuracy at all certification levels, since more accurate and confident predictions give a larger certified radius $r^\infty$.

Second, as we grow the input dimension $k$, the accuracy of ARS remains either stable or increases, whereas that of other baselines goes down, resulting in an increasing gap. For instance, at $\sigma = 1.0$, the gap between ARS and the best baseline is 1.3% points for $k = 32$, 2.5% points for $k = 48$,

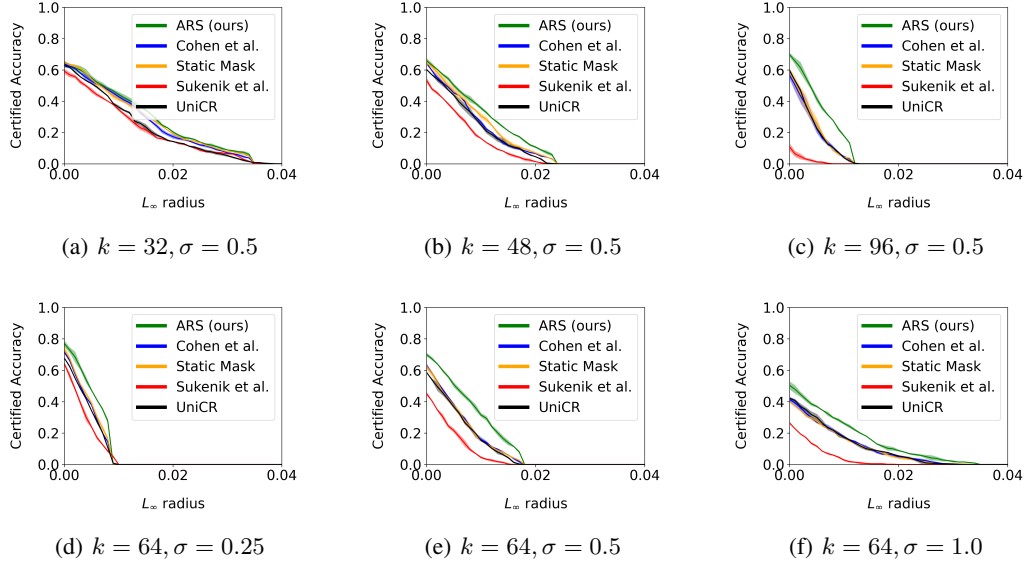

(a) $k = 32, \sigma = 0.5$

(b) $k = 48, \sigma = 0.5$

(c) $k = 96, \sigma = 0.5$

(d) $k = 64, \sigma = 0.25$

(e) $k = 64, \sigma = 0.5$

(f) $k = 64, \sigma = 1.0$

Figure 2: **Certified Test Accuracy on CIFAR-10 (20kBG).** (a)-(c) show the effect of dimensionality for (a) no background / $k = 32$, (b) $k = 48$, and (c) $k = 96$ for constant $\sigma = 0.5$. (d)-(f) show the effect of noise for (d) $\sigma = 0.25$, (e) $\sigma = 0.5$ and (f) $\sigma = 1.0$ with dimensionality fixed to $k = 64$. Each line is the mean and the shaded interval covers +/- one standard deviation across seeds.

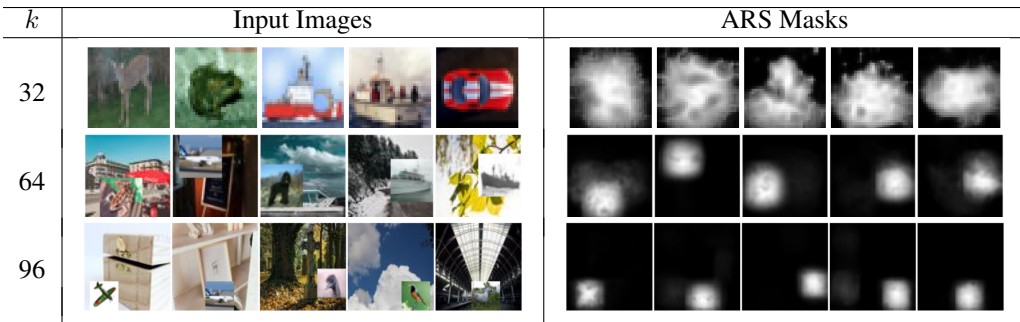

Figure 3: *(left)* Original CIFAR-10 images superimposed on backgrounds for different $k$ (except $k = 32$ which is no background), and *(right)* their corresponding masks (grayscale) inferred by our mask model $w$. All masks are for $\sigma = 0.5$. Appendix D.2 shows all the corresponding images across our multi-step architecture.

8.2% points for $k = 64$ and 15% points for $k = 96$. As $k$ grows, the amount of relevant information (a $32 \times 32 \times 3$ CIFAR-10 image) remains the same, whereas the amount of spurious background information increases. ARS' mask is able to rule out spurious pixels, reducing the noise in the second step (Figure 3). Thanks to this masking, ARS is much less sensitive to increases in dimensionality.

Third, we observe that except $k = 32$, ARS improves over all the baselines in low ($\sigma = 0.25$) to high ($\sigma = 1.0$) noise regimes. In fact, this trend continues to persist in higher noise regimes (Appendix D). Similar to previous observations, we notice that as we increase $\sigma$, other baselines's accuracy drops significantly whereas ARS accuracy drops much less, displaying higher noise tolerance.

ARS training and inference requires additional computation. To certify a single input ($k = 32$), Cohen et al. (2019) takes ~12 seconds while ARS takes ~26 seconds (as measured on an NVIDIA A100 80Gb GPU). This 2× overhead does however yield improved certified accuracy.

## 4.2 CelebA Benchmark: Classification Without Spatial Alignment

To evaluate ARS on a more realistic task with natural spatial variation, we use the CelebA face dataset in its unaligned version. We focus on the "mouth slightly open" (label 21) binary classification task because mouth location and shape vary. The input part relevant to this task is likely well-localized, which affords an opportunity for the mask model to reduce the effective input dimension. The dataset

| Setting/Approach | Cohen et al. | Static Mask | ARS |
|---|---|---|---|
| CelebA, $\sigma = 0.25$ | 94.3 (0.5) | 93.0 (0.8) | **97.0 (0.8)** |
| CelebA, $\sigma = 0.5$ | 91.0 (0.8) | 91.7 (0.9) | **94.3 (0.9)** |
| CelebA, $\sigma = 1.0$ | 83.3 (0.9) | 84.7 (2.6) | **91.0 (1.6)** |

Table 2: **Standard test accuracy ($r = 0$) on CelebA (unaligned and cropped)**. ARS is equal or better. Adaptivity handles the higher spatial dimensions ($160 \times 160$) and variation of these inputs.

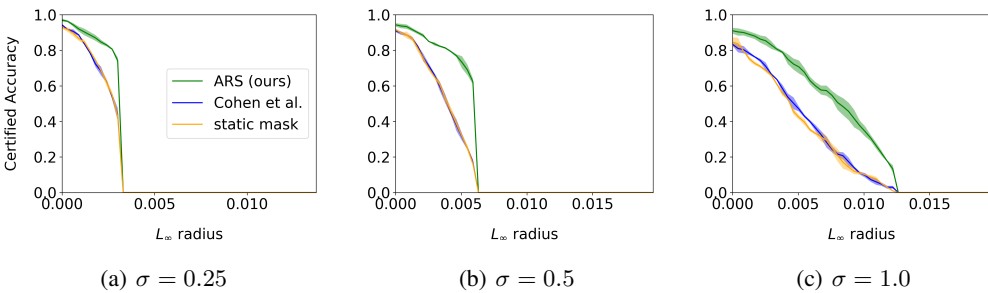

(a) $\sigma = 0.25$        (b) $\sigma = 0.5$        (c) $\sigma = 1.0$

Figure 4: **Certified test accuracy on CelebA (unaligned and cropped).** We evaluate static methods and ARS to measure the value of adaptivity. Each line is the mean and the shading covers $\pm 1$ standard deviation across three seeds. Adaptivity helps at all noise levels.

consists of images with varied resolution, and meta-data about the position of different features, including the mouth. To create a challenging benchmark, we randomly crop all images to $160 \times 160$ pixels, which creates spatial variation in the mouth's position. The only crop constraint is that the mouth is $\geq 10$ pixels from the edge to ensure sufficient input to solve the task. Figure 5 shows example images from the test set, their respective masks from ARS. and the baseline static mask.

Figure 4 shows the certified accuracy for ARS, Cohen et al. (2019), and static mask, for three levels of the noise $\sigma$. First, both baselines perform very similarly. We can see from Figure 5 that the static mask is approximately identity (notice the $\geq 0.99$ scale), with only very slight dimming on the edges. This is because the mouth is not centred in our benchmark, so there is no one-size-fits-all mask. Second, ARS is able to predict a sparse mask that focuses on

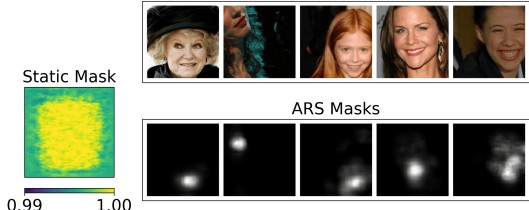

Figure 5: ARS masks are localized and input specific.

areas likely to have the mouth. The mask adapts to each input at test time, which is what enables the sparsity without performance degradation. Third, this sparse mask leads to a large noise reduction, enabling ARS to drastically improve both standard and certified accuracy. For instance, with $\sigma = 0.5$, ARS improves the standard accuracy from $91.0\%$ to $94.3\%$ (a 3.3 point improvement), while the certified accuracy at $r^\infty = 0.004$ jumps from $45.0\%$ to $79.3\%$ (a more than 30 point improvement!). At lower noise ($\sigma = 0.25$) there is still an increase in standard accuracy from $94.3\%$ to $97.0\%$, and an increase in certified accuracy from $68.3\%$ to $84.7\%$ at $r^\infty = 0.002$). At larger noise ($\sigma = 1.0$), ARS sees significant increases ($7.7\%$ points in standard accuracy, and from $21.3\%$ to $49.3\%$ in certified accuracy at $r^\infty = 0.008$).

### 4.3 ImageNet Benchmark: Classification on the Standard Large-Scale Dataset

To evaluate the scalability of ARS we experiment on ImageNet (without any modification) with $\sigma = 0.25, 0.5, 1.0$. For each noise level, we compare with Cohen et al. (2019), which we reproduce for this large-scale setting. We evaluate two versions of ARS: our regular setting (End-To-End); and a version that fixes the base classifier to the model trained as in Cohen et al. (2019), and only trains our mask model for 10 epochs (Pretrain). The certified accuracy is plotted in Figure 6 and the standard accuracy is reported in Table 3.

When only training the mask model, certified accuracy remains close to that of Cohen et al. (2019) at all radii and noise levels. ARS trained end-to-end improves both standard and certified accuracy.

| Setting/Approach | Cohen et al. | ARS (Pretrain) | ARS (End-To-End) |
|---|---|---|---|
| ImageNet, $\sigma = 0.25$ | 66.5 (0.009) | **67.4 (0.002)** | 65.7 (0.006) |
| ImageNet, $\sigma = 0.5$ | 57.2 (0.009) | 56.0 (0.003) | **57.4 (0.010)** |
| ImageNet, $\sigma = 1.0$ | 43.6 (0.005) | 43.8 (0.002) | **44.5 (0.010)** |

Table 3: **Standard test accuracy ($r = 0$) on ImageNet**. ARS maintains standard accuracy.

The standard accuracy (at $r = 0$) increases from $57.2\%$ to $57.4\%$ and from $43.6\%$ to $44.5\%$ when $\sigma = 0.5$ and $\sigma = 1$, respectively. For $\sigma = 0.25$, standard accuracy for ARS is close but slightly lower than Cohen et al. (2019), while the pretrained ARS outperforms Cohen et al. (2019) from $66.5\%$ to $67.4\%$. Appendix F discusses other ARS improvements without certification.

Certified accuracy increases at larger $\sigma$. For instance at $\sigma = 0.5$, ARS improves certified accuracy at $r^\infty = 0.001$ from $48.9\%$ to $50.5\%$. At larger noise $\sigma = 1.0$, ARS improves certified accuracy at $r^\infty = 0.005$ from $21.7\%$ to $23.1\%$. This shows that ARS' adaptivity generalizes outside of the specialized benchmarks we designed, and can scale to large datasets and complex classification tasks.

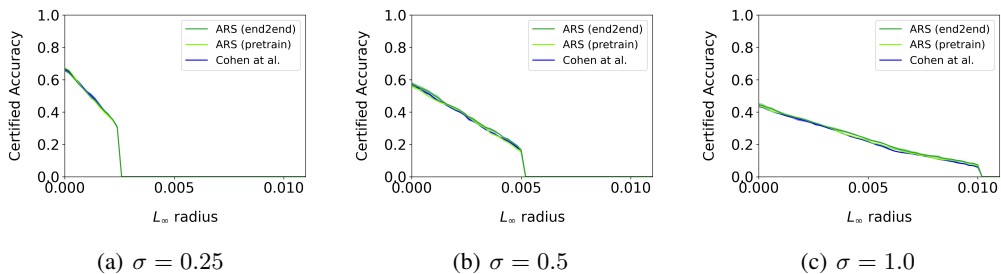

(a) $\sigma = 0.25$      (b) $\sigma = 0.5$      (c) $\sigma = 1.0$

Figure 6: **Certified test accuracy on ImageNet for** $\sigma = 0.25, 0.5, 1$. We plot the mean and the shading covers $\pm 1$ standard deviation for three seeds. ARS is equal or better than non-adaptive RS (Cohen et al.) at large scale.

## 5 Discussion

**Limitations:** The multi-step adaptivity of ARS improves certification at the cost of increased model size and computation for RS. This impacts both training and testing computation, and is especially costly in the context of RS due to the Monte Carlo estimation of the model's expected predictions (over several forward passes at inference time). While we empirically show improvement by ARS, it would be interesting and important to investigate how it combines with other RS improvements such as adversarial training (Salman et al., 2019), consistency regularization (Jeong and Shin, 2020), higher order certification (Mohapatra et al., 2020), double sampling (Li et al., 2022b), and denoising by diffusion (Carlini et al., 2022). Lastly, our adaptive masking technique provides improved certificates for the $L_\infty$ norm, but does not have the same effect for other norms such as $L_2$ (see remarks in §2.4). It is plausible that ARS' adaptivity can lead to improvements under alternative norms, by leveraging different DP mechanisms and updates. We leave this exploration for future work.

**Implications:** Revisiting heuristic adaptive defences (as surveyed in Croce et al. (2022)) through the lens of ARS could help improve the empirical performance of provable defences. ARS may require extensions, but could eventually enable the formal analysis of input purification (e.g., Song and Kim (2018); Nie et al. (2022); Yoon et al. (2021)), or leverage DP-SGD (Abadi et al., 2016) to analyze defences that update by test-time optimization (Alfarra et al., 2022b; Hwang et al., 2022; Mao et al., 2021). Going further, one could leverage the vast DP literature to extend ARS, enabling fully-adaptive variance defences inspired by Alfarra et al. (2022a) by leveraging privacy odometers (Rogers et al., 2016; Lécuyer, 2021; Whitehouse et al., 2023).

To conclude: we introduced Adaptive Randomized Smoothing (ARS) to reconnect RS with DP theory, to propose a new two-step defence for deep image classification, and to rigorously analyze such adaptive defences that condition on inputs at test time. This framework opens promising avenues for designing models that are adaptively and soundly robust with provable guarantees about their updates on natural and adversarial inputs.

## Acknowledgments and Disclosure of Funding

We thank Krishnamurthy Dvijotham and Ian Goodfellow for reviewing and providing feedback during the Google DeepMind publication process. We thank Motasem Alfarra for discussing variations on input-dependent variance for randomized smoothing. We are grateful for the support of the Natural Sciences and Engineering Research Council of Canada (NSERC) [reference number RGPIN-2022-04469], as well as a Google Research Scholar award. This research was enabled by computational support provided by the Digital Research Alliance of Canada (alliancecan.ca).

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

# A  $f$-DP Background

For this background we use the DP mechanism terminology. A mechanism $\mathcal{M}(.)$ is a randomized computation taking an input and returning one sample from the distribution of outputs for this input: $m \sim \mathcal{M}(x)$ with input $x$ and output $m$. In ARS, each model step corresponds to an $f$-DP mechanism.

**Definitions.** Dong et al. (2019; 2022) formalize privacy as a bound on the power of hypothesis tests. Consider any two neighbouring inputs: in the most common DP applications, $X, X'$ are two databases differing in one element; in the case of ARS against an $L_p$ adversary, $X, X' \in \mathbb{R}^d$ are any two inputs such that $X - X' \in B_p(r)$. Intuitively a randomized mechanism $\mathcal{M}$ is private if, for any such neighbouring inputs, the distributions $\mathcal{M}(X)$ and $\mathcal{M}(X')$ are hard to distinguish. That is, by looking at a sample output from mechanism $\mathcal{M}$, it is hard to guess whether $\mathcal{M}$ ran on $X$ or on $X'$.

In $f$-DP (Dong et al., 2019; 2022) "hard to distinguish" is defined by a hypothesis testing problem:

$$\mathcal{H}_0 : \text{the input was } X \qquad \text{vs.} \qquad \mathcal{H}_1 : \text{the input was } X'.$$

The output $m \sim \mathcal{M}$ serves as input to a rejection rule $\phi(.) \in [0,1]$ (note: to preserve typical notations, lower-case $\phi$ is the rejection rule, and upper-case $\Phi$ is the standard normal CDF). The rejection rule rejects $\mathcal{H}_0$ with probability $\phi(m)$, so $\phi(m) = 0$ predicts that $X$ was the input, and $\phi(m) = 1$ that $X'$ was.

Given a rejection rule $\phi$, we define its Type I error $\alpha_\phi$ and type II error (or one minus the power of the rule) $\beta_\phi$ as:

$$\alpha_\phi \triangleq \mathbb{E}_{m \sim \mathcal{M}(X)}[\phi(m)] \qquad \beta_\phi \triangleq 1 - \mathbb{E}_{m \sim \mathcal{M}(X')}[\phi(m)]$$

Intuitively, $\alpha_\phi$ is the expected amount of rejection of $\mathcal{H}_0$ when the hypothesis is correct ($X$ was in input, but we think $X'$ was), also called the level of the rejection rule. On the flip side, $\beta_\phi$ is the expected amount of non-rejection under $\mathcal{H}_1$ ($X'$ was in input, but we think $X$ was). $1 - \beta_\phi$ is called the power of the rejection rule.

For any two distributions $\mathcal{M}(X)$ and $\mathcal{M}(X')$, we define the trade-off function $T\big(\mathcal{M}(X), \mathcal{M}(X')\big) : [0,1] \to [0,1]$ that quantifies the minimum amount of type II error achievable at each value of type I error by any (measurable) rule; or equivalently the maximum power of any rule at each level:

$$\forall \alpha \in [0,1], \ T\big(\mathcal{M}(X), \mathcal{M}(X')\big)(\alpha) = \inf_\phi \{\beta_\phi : \alpha_\phi \leq \alpha\}$$

Now we define $f$-DP: for any trade-off function $f$, a mechanism $\mathcal{M}$ is $f$-DP if, for any neighbouring inputs $X, X'$,

$$T\big(\mathcal{M}(X), \mathcal{M}(X')\big) \geq f$$

These definitions are the main technical tools we need to prove Proposition 2.1. Corollary 2.2 only adds the formula for $f$ for the Gaussian mechanism, given in Section 2.1.

**Composition.** All other results rely on the above plus the adaptive composition of $f$-DP mechanisms. Such composition is key to all DP theory and algorithm design. Consider a sequence of $N$ mechanisms $\mathcal{M}_i$, such that each mechanism is $f_i$-DP with regards to $X, X'$, and depends on the neighbouring input as well as the output of all previous mechanisms. More formally, under $\mathcal{H}_0$ we have $m_i \sim \mathcal{M}_i(X, m_{<i})$, and under $\mathcal{H}_1$ we have $m_i \sim \mathcal{M}_i(X, m_{<i})$, where $m_{<i} \triangleq (m_1, \ldots, m_{i-1})$. Concretely, each $\mathcal{M}_i$ is $f_i$-DP with regards to $X, X'$ for $f_i$ known in advance, but the actual computation made by $\mathcal{M}_i$ can depend on $m_{<i}$ (as long as it is $f_i$-DP). We leverage this adaptivity to lower the noise variance in our method's second step while keeping $f_2$ fixed (see §3).

We need two more results to define the composition of a sequence of mechanisms. First, Proposition 2.2 in Dong et al. (2019; 2022) shows that for any trade-off function $f$, there exist two distributions $P_f, Q_f$ such that $T(P_f, Q_f) = f$. Call any such pair of distributions a representative pair of $f$. Second, we define the composition operator $\otimes$ by $f \otimes g = T(P_f \times P_g, Q_f \times Q_g)$. That is, the composition operator between two trade-off functions is the trade-off function between the product distributions on their representative pair. Then Theorem 3.2 in Dong et al. (2019; 2022) shows that:

$$\mathcal{M} : X \to (\mathcal{M}_1(X), \ldots, \mathcal{M}_N(X, y_{<i})) \text{ is } f_1 \otimes \ldots \otimes f_N\text{-DP}.$$

Concretely, the mechanism that returns the sequence of results for all compute adaptive $\mathcal{M}_N$ is $f_1 \otimes \ldots \otimes f_N$-DP. The previous definitions, as well as this composition result, is what we use to prove Theorem 2.3 and Proposition 2.4.

# B Proofs

**Proposition 2.1** (*f*-DP Robustness). *Let $\mathcal{M} : \mathcal{X} \to \mathcal{Y}$ be $f$-DP for $B_p(r)$ neighbourhoods, and let $M_S : X \to \arg\max_{y \in \mathcal{Y}} \mathbb{P}(\mathcal{M}(X) = y)$ be the associated smooth classifier. Let $y_+ \triangleq M_S(X)$ be the prediction on input $X$, and let $\underline{p_+}, \overline{p_-} \in [0,1]$ be such that $\mathbb{P}(\mathcal{M}(X) = y_+) \geq \underline{p_+} \geq \overline{p_-} \geq \max_{y_- \neq y_+} \mathbb{P}(\mathcal{M}(X) = y_-)$. Then:*

$$f(1 - \underline{p_+}) \geq 1 - f(\overline{p_-}) \Rightarrow \forall e \in B_p(r),\ M_S(X + e) = y_+$$

*Proof.* Let us first consider any runner-up class $y_-$. Calling $M$ the random variable for $\mathcal{M}$'s prediction, consider the rejection rule $\phi = \mathbb{1}\{M = y_-\}$, where $\mathbb{1}$ is the indicator function. Denoting $\alpha \triangleq \mathbb{E}_{\mathcal{M}(X)}(\phi)$, and using the fact that $\mathcal{M}$ is $f$-DP for $B_p(r)$ neighbourhoods, we have that $\forall e \in B_p(r)$:

$$\begin{aligned}
\mathbb{P}(\mathcal{M}(X + e) = y_-) &= \mathbb{E}_{\mathcal{M}(X+e)}(\phi) \\
&\leq 1 - f(\alpha) \leq 1 - f(\overline{p_-}),
\end{aligned} \tag{8}$$

where the last inequality is because $\alpha = \mathbb{E}_{\mathcal{M}(X)}(\phi) = \mathbb{P}(\mathcal{M}(X) = y_-) \leq \overline{p_-}$, and $f$ is non-increasing so $f(\alpha) \geq f(\overline{p_-})$ and hence $1 - f(\alpha) \leq 1 - f(\overline{p_-})$.

Let us now consider the predicted class $y_+$. Keeping the same notations, and defining the rule $\phi' = \mathbb{1}\{M \neq y_+\} = 1 - \mathbb{1}\{M = y_+\}$. Then $\alpha' = \mathbb{E}_{\mathcal{M}(X)}(\phi') = 1 - \mathbb{P}(\mathcal{M}(X) = y_+) \leq 1 - \underline{p_+}$, and $\mathbb{E}_{\mathcal{M}(X+e)}(\phi') \leq 1 - f(\alpha') \leq 1 - f(1 - \underline{p_+})$, yielding:

$$\begin{aligned}
\mathbb{P}(\mathcal{M}(X + e) = y_+) &= 1 - \mathbb{E}_{\mathcal{M}(X+e)}(\phi') \\
&\geq f(1 - \underline{p_+}).
\end{aligned} \tag{9}$$

Putting Equations (8) and (9) together, we have that $\mathbb{P}(\mathcal{M}(X+e) = y_+) \geq f(1 - \underline{p_+}) \geq 1 - f(\overline{p_-}) \geq \mathbb{P}(\mathcal{M}(X + e) = y_-)$ and thus $m_S(X + e) = y_+$. $\square$

Note that we do not have to chose a rule $\phi \in \{0, 1\}$, but could instead return any number in $[0, 1]$, such as the logits of the base classification model, yielding the following definition for the smoothed classifier $M_S : X \to \arg\max_{y \in \mathcal{Y}} \mathbb{E}(\mathcal{M}(X)_y)$.

**Proposition 2.2** (RS from *f*-DP). *Let $\mathcal{M} : X \to g(X + z)$, $z \sim \mathcal{N}(0, \sigma^2\mathbb{I}^d)$, and $M_S : X \to \arg\max_{y \in \mathcal{Y}} \mathbb{P}(\mathcal{M}(X) = y)$ be the associated smooth model. Let $y_+ \triangleq M_S(X)$ be the prediction on input $X$, and let $\underline{p_+}, \overline{p_-} \in [0,1]$ be such that $\mathbb{P}(\mathcal{M}(X) = y_+) \geq \underline{p_+} \geq \overline{p_-} \geq \max_{y_- \neq y_+} \mathbb{P}(\mathcal{M}(X) = y_-)$. Then $\forall e \in B_2(r_x),\ M_S(X + e) = y_+$, with:*

$$r_X = \frac{\sigma}{2}\left(\Phi^{-1}(\underline{p_+}) - \Phi^{-1}(\overline{p_-})\right).$$

*Proof.* $X :\to X + z$, $z \sim \mathcal{N}(0, \sigma^2)$ is a Gaussian mechanism. By Equation (2), for the $B_r(r)$ neighbouring definition, it is $G_{\frac{r}{\sigma}}$-DP. By post-processing $\mathcal{M}$ is also $G_{\frac{r}{\sigma}}$-DP.

Applying Proposition 2.1, we have that $G_{\frac{r}{\sigma}}(1 - \underline{p_+}) \geq 1 - G_{\frac{r}{\sigma}}(\overline{p_-}) \Rightarrow \forall e \in B_2(r),\ m_S(X + e) = y_+$. Let us find $r_X = \sup\{r : G_{\frac{r}{\sigma}}(1 - \underline{p_+}) \geq 1 - G_{\frac{r}{\sigma}}(\overline{p_-})\}$. Since $G_{\frac{r}{\sigma}}(.)$ as a function of $r$ is monotonously decreasing this will happen at $G_{\frac{r_X}{\sigma}}(1 - \underline{p_+}) = 1 - G_{\frac{r_X}{\sigma}}(\overline{p_-})$, that is:

$$\begin{aligned}
& \Phi\left(\Phi^{-1}(\underline{p_+}) - \frac{r_X}{\sigma}\right) = 1 - \Phi\left(\Phi^{-1}(1 - \overline{p_-}) - \frac{r_X}{\sigma}\right) \\
\Rightarrow\ & \Phi^{-1}(\underline{p_+}) - \frac{r_X}{\sigma} = -\Phi^{-1}(1 - \overline{p_-}) + \frac{r_X}{\sigma} \\
\Rightarrow\ & \Phi^{-1}(\underline{p_+}) - \frac{r_X}{\sigma} = \Phi^{-1}(\overline{p_-}) + \frac{r_X}{\sigma} \\
\Rightarrow\ & r_X = \frac{\sigma}{2}\left(\Phi^{-1}(\underline{p_+}) - \Phi^{-1}(\overline{p_-})\right),
\end{aligned}$$

where the first implication holds because by symmetry of the standard normal $1 - \Phi(x) = \Phi(-x)$, and because $\Phi$ is strictly monotonous ; the second because similarly, $\Phi^{-1}(1 - p) = -\Phi^{-1}(p)$. $\square$

## C  Experiment Details

### C.1  Mask Architecture

Figure 7 shows the architecture of our Mask model $w$ ($\mathcal{M}_1$). We adapt a UNet architecture to preserve dimensions, and use a Sigmoid layer at the end of the model to output values between 0 and 1 for mask weights. We set up our UNet hyperparameters as : *in_channels*=3, *out_channels*=1 (to out put a mask), *base_channel*=32, *channel_mult*={1,2,4,8}.

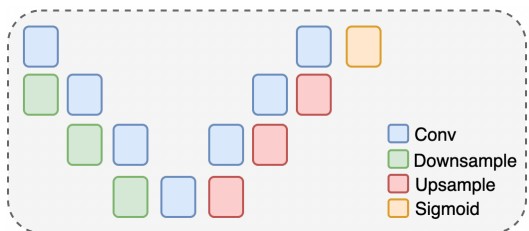

Figure 7: UNet structure

| | | CIFAR-10 | CelebA | ImageNet |
|---|---|---|---|---|
| | GPU | single 24G RTX4090 | single 24G RTX4090 | single 80G A100 |
| | epoch | 100 | 24 | 100(10+90) |
| | train batch size | 256 | 64 | 300 |
| | test batch size | 20 | 100 | 100 |
| | base channel | 32 | 64 | 16 |
| | optimizer | AdamW | SGD | SGD |
| Mask | lr | 1e-3 | 5e-2 | 5e-2 |
| Model | weight decay | 1e-4 | - | 1e-4 |
| (UNet) | momentum | 0.9 | - | 0.9 |
| | step size | 40 | - | 30 |
| | gamma | 0.5 | - | 0.1 |
| | model | ResNet110 | ResNet50 | ResNet50 |
| | optimizer | AdamW | SGD | SGD |
| Base | lr | 1e-2 | 5e-2 | 1e-1 |
| Classifier | weight decay | 1e-4 | - | 1e-4 |
| | momentum | 0.9 | - | 0.9 |
| | step size | 30 | 3 | 30 |
| | gamma | 0.1 | 0.8 | 0.1 |

Table 4: Hyperparameters for training ARS. Check Appendix C.3 for more details of CIFAR-10 hyperparameters.

### C.2  Hyperparameter tuning for CelebA

Table 4 provides the details of our ARS models' hyper-parameters. On the CelebA dataset, we tune the hyper-parameters in the ARS, Cohen et al. (2019), and static mask settings at $\sigma = 0.75$. In all settings, we settle on SGD with learning rate 0.05 and a step learning rate scheduler (step size of 3 and $\gamma = 0.8$) for the base classifier. In the static mask setting, we use SGD with learning rate 0.01.

### C.3  Hyperparameter tuning for CIFAR-10 BG20k

For the base classifier $g$ in Cohen et al. (2019), Static Mask and ARS experiments, we tune the optimizer and its hyperparameters using $k = 64$, $\sigma = 1.5$ as the testbed. Based on these tuning experiments, we chose AdamW as the optimizer with an initial learning rate of 0.01 and weight decay to 0.0001. We scale the learning rate by 0.1 every $30^{th}$ epoch, with a batch size of 256. For the mask model $w$ in ARS experiments, we again used the AdamW optimizer with an initial learning rate of 0.001, weight decay of 0.0001, whilst scaling the learning rate by 0.5 every $40^{th}$ epoch. We used the same hyperparameters for all $k$'s and $\sigma$'s for these setups.

For Súkeník et al. (2021), for $k = 32, 48, 64$, we tune the hyperparameters using $k = 64$ as the testbed. We kept the optimizer same as used in the author's code (SGD), setting the initial learning rate to 0.01, momentum to 0.9 and weight decay to 0. We scaled the learning rate by 0.1 every $30^{th}$ epoch. We tune $k = 96$'s parameters separately. We start with an initial learning rate 0.1, keeping

the rest of hyperparameters same as for $k = 32, 48, 64$. Note that for $k = 96$, we could not get the standard accuracy to improve upon random baseline for $\sigma = \{0.5, 0.75, 1.0, 1.5\}$, despite extensively tuning the learning rate.

| $k$ | $\sigma = 0.12$ | $\sigma = 0.25$ | $\sigma = 0.5$ | $\sigma = 0.75$ | $\sigma = 1.0$ | $\sigma = 1.5$ |
|---|---|---|---|---|---|---|
| 32 | 2.25 | 2.25 | 2.25 | 2.25 | 2.25 | 2.25 |
| 48 | 2.0 | 2.0 | 2.25 | 2.0 | 2.25 | 2.0 |
| 64 | 2.25 | 2.0 | 2.25 | 2.0 | 2.0 | 2.0 |
| 96 | 2.0 | 2.0 | 2.25 | 2.25 | 2.25 | 2.25 |

Table 5: UniCR $\beta$ chosen for each $k$, $\sigma$ setting.

For UniCR, we tune $\beta$ (the parameter of the generalized normal distribution for the noise) using $\sigma = 0.75$ and $k = 48$ as the testbed. We perform grid search on $\beta$ and find that $\beta = 2.0$ or $\beta = 2.25$ (Gaussian and close to a Gaussian, but with a wider more and shorted tails) perform best. For each $k$, $\sigma$ setting, we train 3 models with $\beta = 2.25$ and $2.0$ and choose the $\beta$ giving highest mean standard accuracy. The chosen $\beta$ for each setting is given in Table 5. For each setting we also tune optimizer hyper-parameters. At $k = 32$, we use SGD optimizer. We use a learning rate of 0.01, momentum of 0.9 and weight decay of 0.0005 with a step learning rate scheduler (30 step and $\gamma = 0.1$). At $k = 48, 64, 96$ we use training batch size 256, 100 epochs, and AdamW optimizer. We use a learning rate of 0.001 with step learning rate scheduler (30 step size and $\gamma = 0.5$).

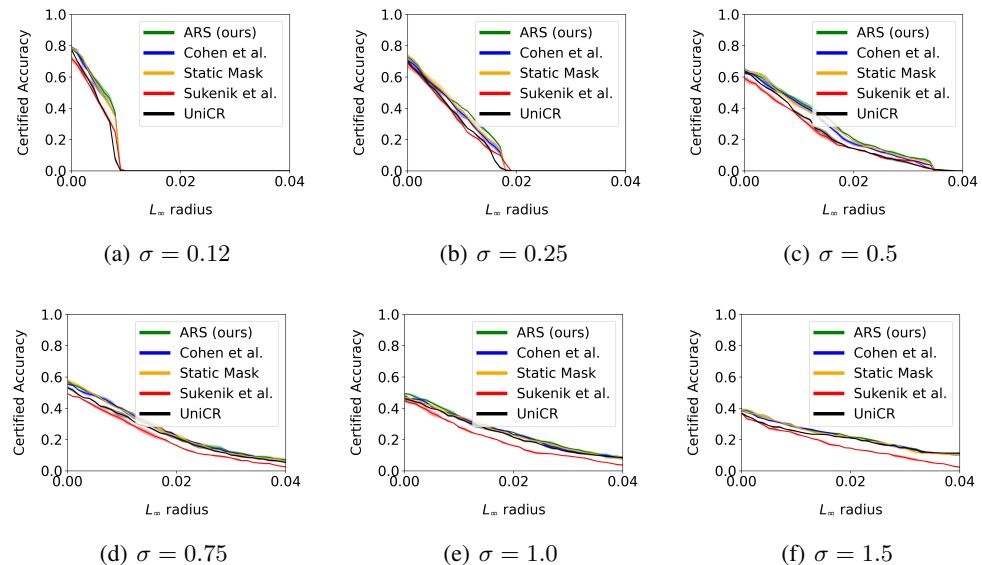

Figure 8: $k = 32$ **certified test accuracy results for CIFAR-10 (20kBG)** (a)-(f) show the effect of increasing $\sigma$. These results are in our 20kBG setting where= a CIFAR-10 image is placed randomly along the edges of a background image. Each line is the mean and the shaded interval covers +/- one standard deviation across seeds.

# D  Additional Results on CIFAR10 BG20k Benchmark

We show here full sweep of results for CIFAR-10 Bg20k benchmark for $k = 32, 48, 64, 96$ and $\sigma = 0.12, 0.25, 0.5, 0.75, 1.0, 1.5$ for all our baselines and ARS. Similar to results presented in Table 1, we report the mean accuracy and standard deviation over three seeds.

| $k$ | $\sigma$ | Cohen et al. | Static Mask | UniCR | Súkeník et al.$^\triangle$ | ARS $^\triangle$ |
|---|---|---|---|---|---|---|
| | 0.12 | **79 (0.7)** | 78.6 (0.9) | 77.5 (1.) | 71.8 (1.6) | 78.5 (0.7) |
| | 0.25 | 70.6 (1.) | **73.9 (0.8)** | 69.8 (1.4) | 68.6 (2.8) | 72.6 (0.9) |
| 32 | 0.5 | 63.6 (2.) | **64.8 (0.9)** | 62.8 (0.8) | 59.1 (1.6) | 64 (1.4) |
| | 0.75 | 55.5 (0.8) | **57.8 (0.8)** | 53.1 (0.8) | 49.3 (0.2) | 57.3 (0.6) |
| | 1.0 | 48 (0.7) | 47.3 (1.3) | 46.1 (0.9) | 44.6 (1.0) | **49.3 (0.6)** |
| | 1.5 | 38.6 (0.2) | **39.1 (0.8)** | 36.8 (0.8) | 36.9 (0.1) | 38.3 (1.0) |
| | 0.12 | 80.1 (0.8) | 80.9 (0.7) | 77.0 (2.9) | 73 (0.1) | **83.6 (0.4)** |
| | 0.25 | 71.6 (1.0) | 72 (2.0) | 69.7 (0.8) | 65 (0.07) | **75.5 (1.0)** |
| 48 | 0.5 | 64.3 (0.2) | 64.1 (1.6) | 60.3 (0.6) | 53.5 (1.8) | **66.0 (0.8)** |
| | 0.75 | 52.5 (1.2) | 56 (0.8) | 52.7 (0.6) | 41.8 (2.4) | **57.6 (1.5)** |
| | 1.0 | 42.5 (2.1) | 45.1 (1.2) | 44.3 (0.2) | 34.1 (1.0) | **47.6 (2.1)** |
| | 1.5 | 30.8 (0.2) | **34.9 (2.4)** | 32.3 (0.2) | 25.5 (1.0) | 34.1 (2.9) |
| | 0.12 | 80.9 (1.0) | 81.3 (1.0) | 76.7 (0.8) | 73.8 (0.2) | **82.3 (1.0)** |
| | 0.25 | 71.67 (0.9) | 73.1 (3.2) | 67.8 (0.5) | 64.1 (0.8) | **77 (1.8)** |
| 64 | 0.5 | 61.6 (2.7) | 64 (1.4) | 58.7 (0.2) | 45.1 (1.1) | **65.4 (1.8)** |
| | 0.75 | 49.4 (2.) | 51.6 (1.0) | 50.7 (0.2) | 31.3 (2.0) | **56.5 (2.6)** |
| | 1.0 | 41.3 (1.8) | 40 (0.5) | 42.2 (0.6) | 26.5 (0.7) | **50.4 (2.5)** |
| | 1.5 | 28.3 (0.2) | 25.9 (3.2) | 30.0 (1.8) | 15.8 (2.7) | **30.6 (1.0)** |
| | 0.12 | 79.1 (0.8) | 79.6 (0.4) | 77.5 (1.1) | 66 (0.7) | **80.3 (3.7)** |
| | 0.25 | 65.3 (1.6) | 71.8 (1.3) | 68.8 (1.8) | 45.5 (0.9) | **78.3 (2.2)** |
| 96 | 0.5 | 56.6 (2.4) | 59.5 (1.4) | 59.7 (1.3) | 10.8 (2.3) | **69.8 (1.2)** |
| | 0.75 | 45.3 (3.4) | 48 (4.3) | 50.2 (0.5) | 10.2 (0.4) | **63.6 (3.5)** |
| | 1.0 | 33.8 (3.8) | 36.9 (0.5) | 41.3 (2.4) | 10.4 (0.3) | **56.3 (2.3)** |
| | 1.5 | 26 (1.4) | 25.1 (0.6) | 31.0 (1.4) | 10.8 (1.4) | **39.1 (2.5)** |

Table 6: **Standard Accuracy** ($r = 0$) **on CIFAR-10 (20kBG).** Our 20kBG benchmark places CIFAR-10 images on larger background images. We report the mean accuracy and standard deviation over three seeds. ARS achieves higher accuracy across noise $\sigma$ and input dimension $k$. $^\triangle$ indicates adaptivity.

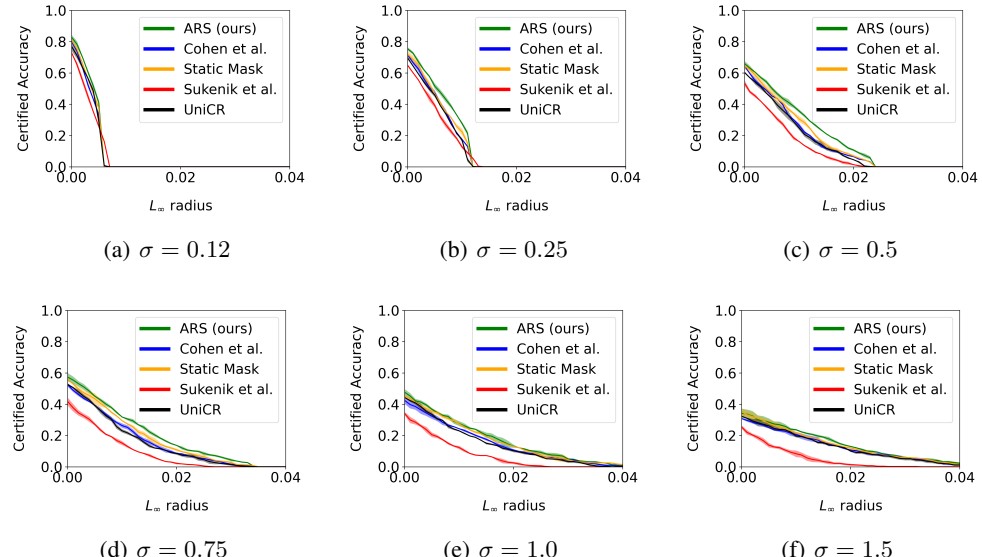

Figure 9: $k = 48$ **certified test accuracy results for CIFAR-10 (20kBG)** (a)-(f) show the effect of increasing $\sigma$. These results are in our 20kBG setting where a CIFAR-10 image is placed randomly along the edges of a background image. Each line is the mean and the shaded interval covers +/- one standard deviation across seeds.

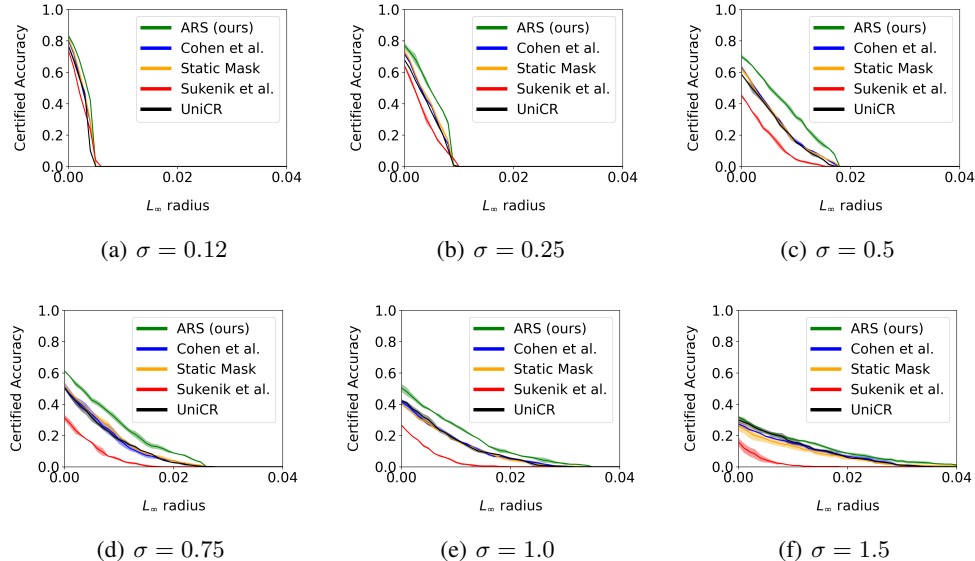

Figure 10: $k = 64$ **certified test accuracy results for CIFAR-10 (20kBG)** (a)-(f) show the effect of increasing $\sigma$. These results are in our 20kBG setting where a CIFAR-10 image is placed randomly along the edges of a background image. Each line is the mean and the shaded interval covers +/- one standard deviation across seeds.

## D.1 Impact of Certification Test-set Size

The baseline results in Table 6 are lower than those reported by Cohen et al. (2019) (Figure 6). This is a result of our certifying over a smaller subset of the test set (the released code from Cohen et al. (2019)[1] uses a subset of size 500, while the paper says the certification was on the full test set). In Table 7 we report the standard accuracy ($r = 0$) on the same 500 samples subset of the CIFAR-10 test set as used in the code released by Cohen et al. (2019). We show results for $k = 32$, which is the plain CIFAR-10 task, using both the hyper-parameters from Cohen et al. (2019), our own optimized hyper-parameters (the most notable change is that we use AdamW), and ARS. We make three observations. First, as it turns out, our 200 samples subset used for results in Table 6

---

[1] https://github.com/locuslab/smoothing/blob/78a4d949e4f627d000a78908e001f8ca66c92943/experiments.MD

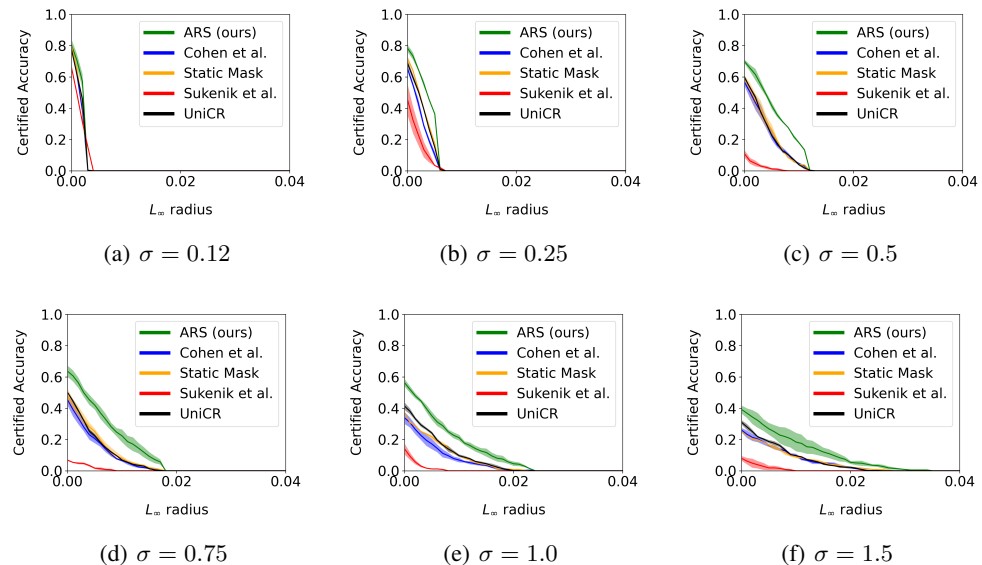

| | (a) $\sigma = 0.12$ | (b) $\sigma = 0.25$ | (c) $\sigma = 0.5$ |
| | (d) $\sigma = 0.75$ | (e) $\sigma = 1.0$ | (f) $\sigma = 1.5$ |

Figure 11: $k = 96$ **certified test accuracy results for CIFAR-10 (20kBG)** (a)-(f) show the effect of increasing $\sigma$. These results are in our 20kBG setting where a CIFAR-10 image is placed randomly along the edges of a background image. Each line is the mean and the shaded interval covers +/- one standard deviation across seeds.

| $\sigma$ | Samples | Approach | | |
|---|---|---|---|---|
| | | Cohen et al. | Cohen et al. w/ AdamW | ARS |
| | 200 | 77.2 (2.7) | 79.0 (0.7) | 78.5 (0.7) |
| 0.12 | 500 | 79.6 (2.1) | 82.1 (0.4) | 82.3 (1.1) |
| | 10k | 80.3 (0.6) | 82.7 (0.6) | 83.4 (0.5) |
| | 200 | 70.7 (3.7) | 70.6 (1.0) | 72.6 (0.9) |
| 0.25 | 500 | 72.4 (2.6) | 75.6 (0.9) | 75.9 (0.9) |
| | 10k | 73.7 (2.6) | 77.2 (0.2) | 77.7 (0.1) |
| | 200 | 62.0 (0.4) | 63.6 (2.0) | 64.0 (1.4) |
| 0.5 | 500 | 63.1 (0.7) | 65.2 (0.9) | 65.9 (0.7) |
| | 10k | 64.3 (0.5) | 66.1 (0.3) | 67.9 (0.3) |
| | 200 | 45.0 (1.5) | 48.0 (0.7) | 49.3 (0.6) |
| 1.0 | 500 | 45.9 (1.6) | 49.1 (0.4) | 50.5 (0.8) |
| | 10k | 46.8 (1.7) | 50.0 (0.5) | 51.5 (0.4) |

Table 7: $k = 32$ **standard accuracy ($r = 0$) on CIFAR-10 (20kBG).** We report the mean accuracy and standard deviation over three seeds. We compare three approaches: Cohen et al. (2019) trained with the hyper-parameters they report in their GitHub repository, Cohen et al. (2019) trained with the hyper-parameters we report in 4, and ARS. We train models with each approach at $\sigma = 0.12, 0.25, 0.5, 1.0$. The corresponding models are certified on 200, 500, and 10,000 samples.

is more challenging, and accuracy values are systematically lower (by about $2\%$ points) than over 500 samples and the full test set (10k samples). Using the same 500 samples subset yields accuracy values very close to those reported by Cohen et al. (2019) on their hyper-parameters. Second, over all samples sizes 200, 500, and 10k, our hyper-parameters significantly improve the accuracy of RS (by about $3\%$ points consistently), confirming that we are making a fair comparison between the best RS models we could find and ARS. Third, at $k = 32$ ARS provides modest improvements over tuned RS, as CIFAR-10 images are well cropped and low-dimensional, providing less opportunity for dimension reduction through masking, and hence lower ARS improvements.

### D.2 CIFAR-10 BG20k figures

Figure 12, Figure 13, Figure 14 and Figure 15 shows figures of different stages in our ARS architecture (Figure 1). In all of these figures, $1^{st}$ row corresponds to input images $X$, $2^{nd}$ row corresponds to images right after $\mathcal{M}_1$, $3^{rd}$ row corresponds to ARS masks, $4^{th}$ row corresponds to element-wise

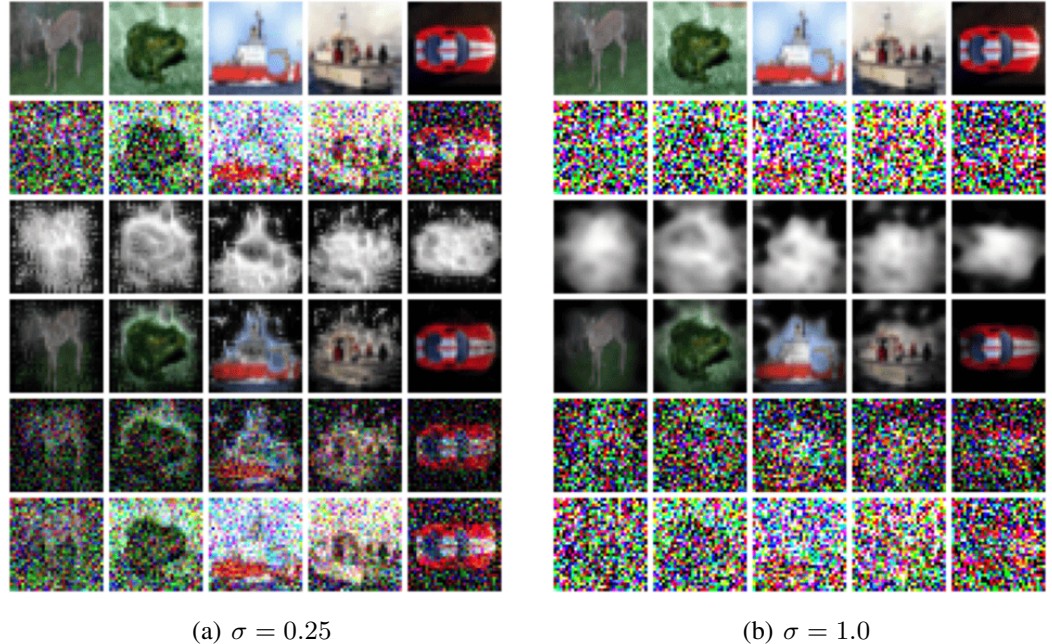

(a) $\sigma = 0.25$                    (b) $\sigma = 1.0$

Figure 12: **Figures at different stages in our ARS architecture for CIFAR-10 $k = 32$ input images. Check Appendix D.2 for detailed information about each row**

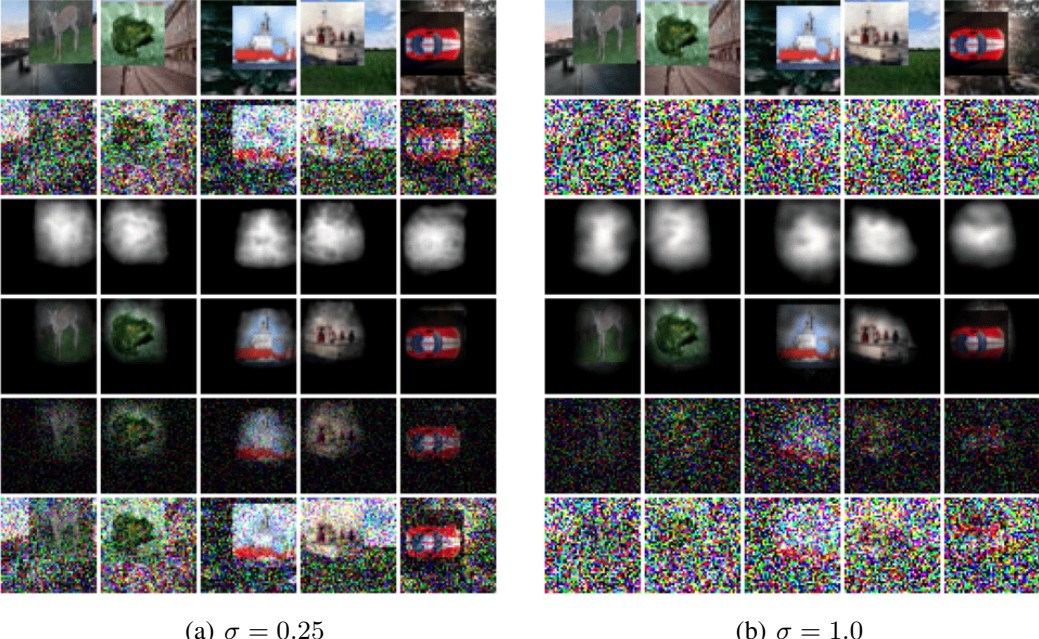

(a) $\sigma = 0.25$                    (b) $\sigma = 1.0$

Figure 13: **Figures at different stages in our ARS architecture for CIFAR-10 BG20k $k = 48$ input images. Check Appendix D.2 for detailed information about each row**

product of input $X$ and ARS masks, $5^{th}$ row corresponds to images right after $\mathcal{M}_2$ and $6^{th}$ row corresponds to images right after averaging $\mathcal{M}_1, \mathcal{M}_2$.

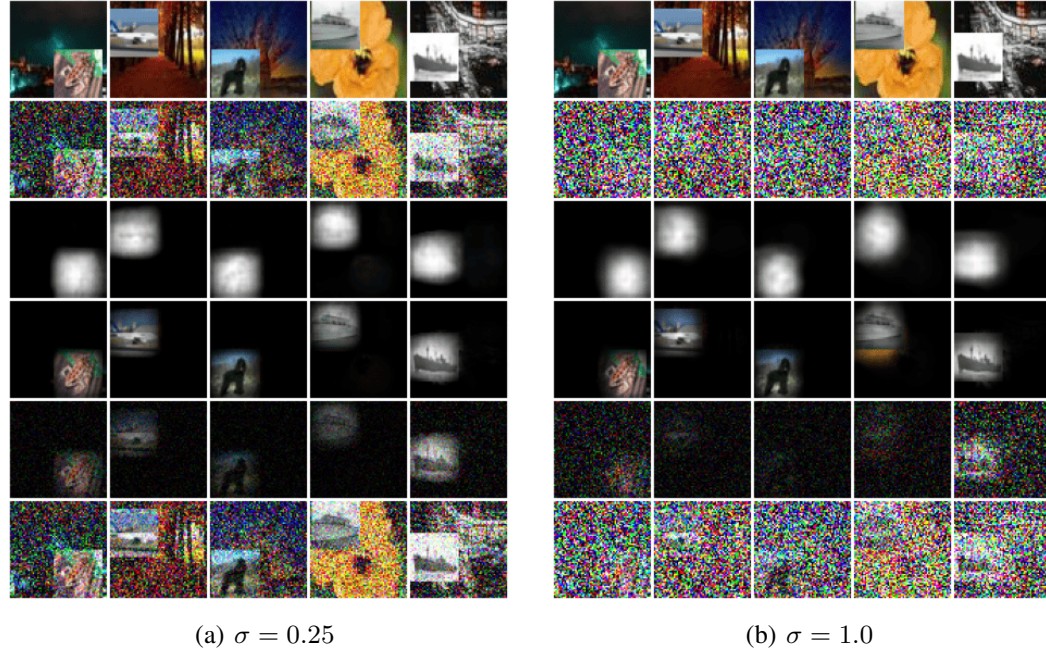

(a) $\sigma = 0.25$        (b) $\sigma = 1.0$

Figure 14: **Figures at different stages in our ARS architecture for CIFAR-10 BG20k** $k = 64$ **input images. Check Appendix D.2 for detailed information about each row**

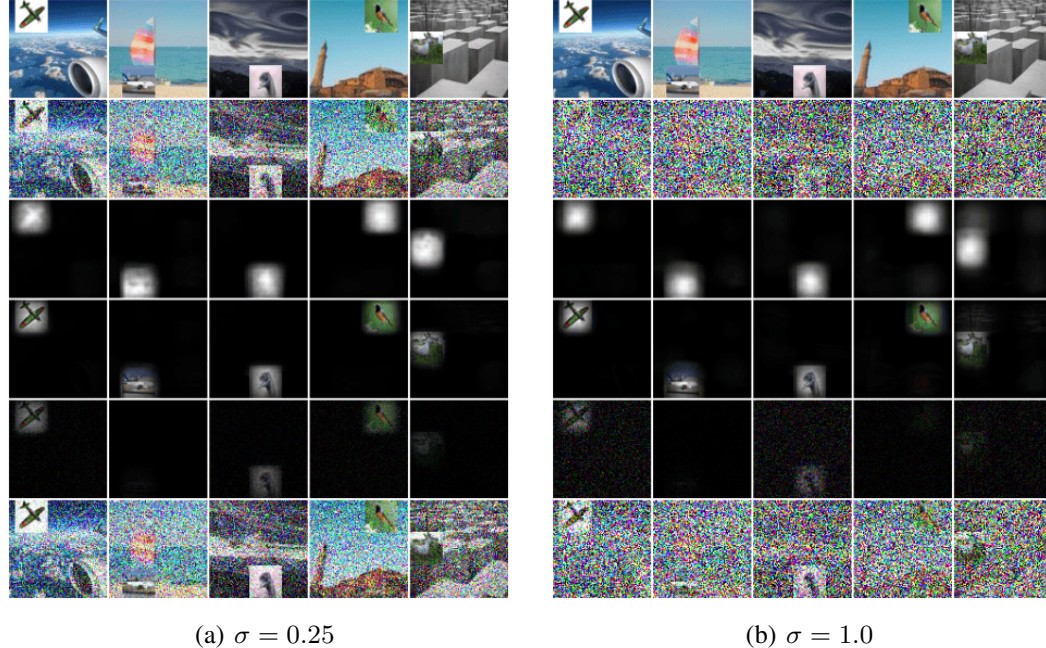

(a) $\sigma = 0.25$        (b) $\sigma = 1.0$

Figure 15: **Figures at different stages in our ARS architecture for CIFAR-10 BG20k** $k = 96$ **input images. Check Appendix D.2 for detailed information about each row**

# E   Additional Results on CelebA

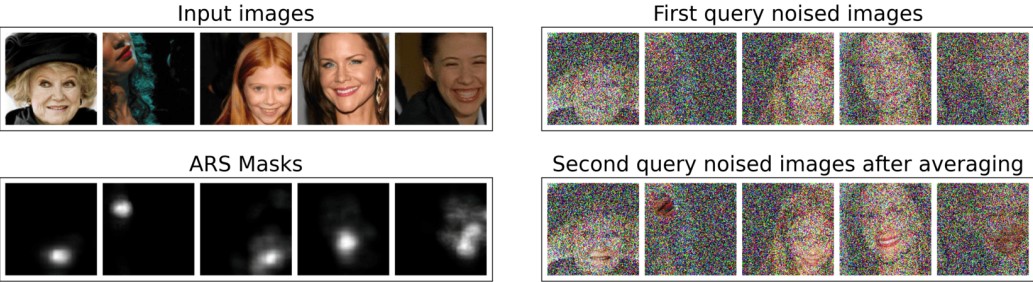

Figure 16: The localized ARS masks produce un-noised mouth regions after averaging.

Figure 16 shows how adaptive masking reduces the noise around areas that are important to classification. The images follow our architecture visualized Figure 1. The mask model is provided the first query noised images as input. The learned masks, presented in the bottom left, are sparse and highly concentrated around the area of interest—the mouth area. The second query noised images (after weighted average) use the mask to clearly reduce the noise around the mouth. This large noise reduction enables ARS to outperform static masking and Cohen et al. (2019), as shown on Figure 4.

We show here a full set of results for the CelebA benchmark over all $\sigma = 0.12, 0.25, 0.5, 0.75, 1.0, 1.5$ for Cohen et al. (2019), static masking, and ARS. Similarly to the results in Table 2, we report the mean standard accuracy and standard deviation over three seeds in Table 8. The certified accuracies are plotted in Figure 17. We see that at all $\sigma$, ARS has the highest standard accuracy.

| $\sigma$ | Cohen et al. | Static Mask | ARS |
|---|---|---|---|
| 0.12 | 94.7 (0.5) | 95.7 (1.2) | **96.3 (0.9)** |
| 0.25 | 94.3 (0.5) | 93.0 (0.8) | **97.0 (0.8)** |
| 0.5 | 91.0 (0.8) | 91.7 (0.9) | **94.3 (0.9)** |
| 0.75 | 88.7 (1.2) | 89.0 (1.4) | **92.3 (0.9)** |
| 1.0 | 83.3 (0.9) | 84.7 (2.6) | **91.0 (1.6)** |
| 1.5 | 77.7 (2.5) | 74.3 (1.7) | **81.0 (2.2)** |

Table 8: **Standard Accuracy ($r = 0$) on CelebA (unaligned and cropped).** We report the mean accuracy and standard deviation over three seeds.

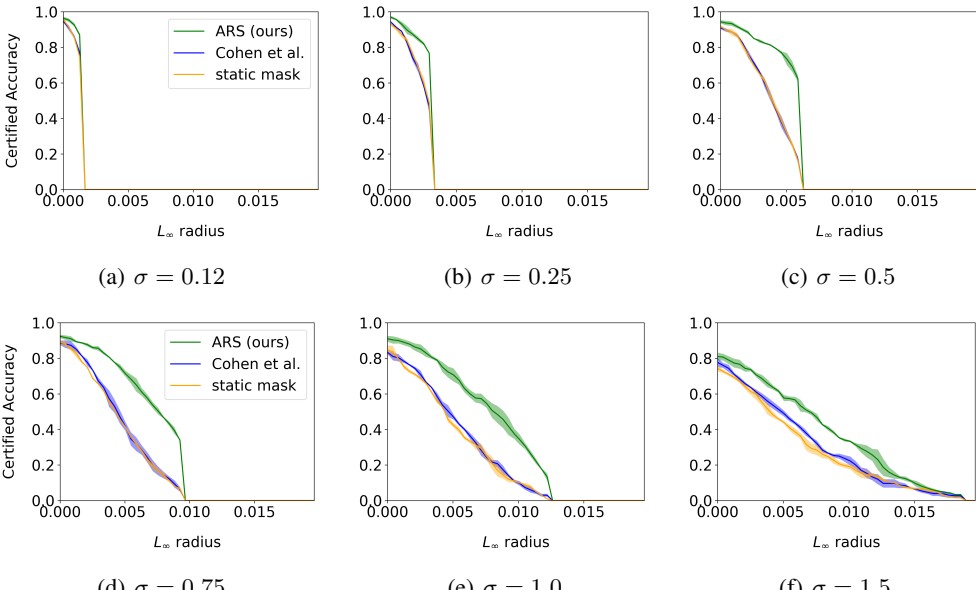

Figure 17: **Certified test accuracy on CelebA (unaligned and cropped).** Each line is the mean and the shading covers $\pm 1$ standard deviation across three seeds. Adaptivity helps at all noise levels.

# F   Additional Results on ImageNet

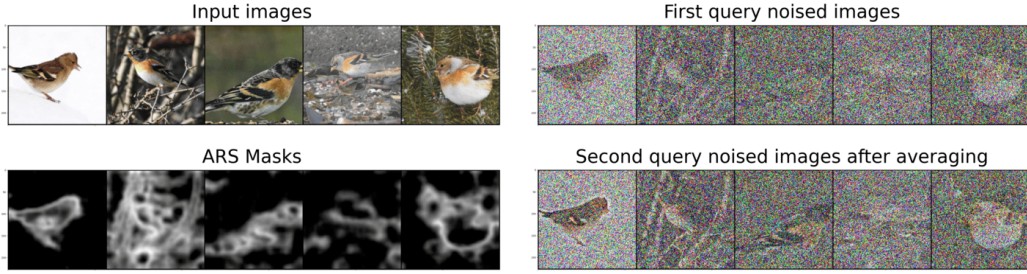

Figure 18: The localized ARS masks produce un-noised object regions after averaging. For $\sigma = 1$.

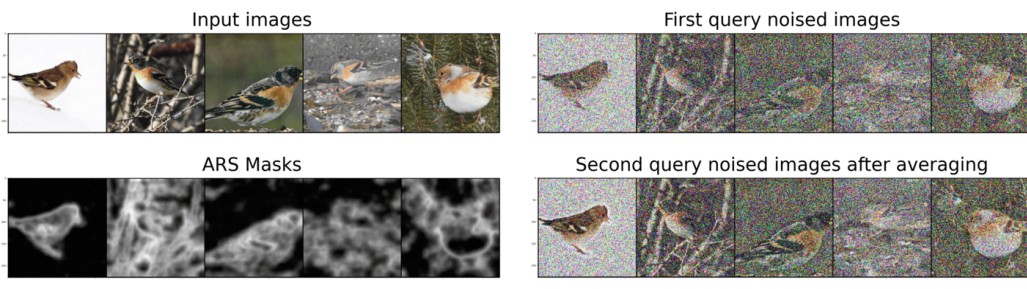

Figure 19: For $\sigma = 0.5$.

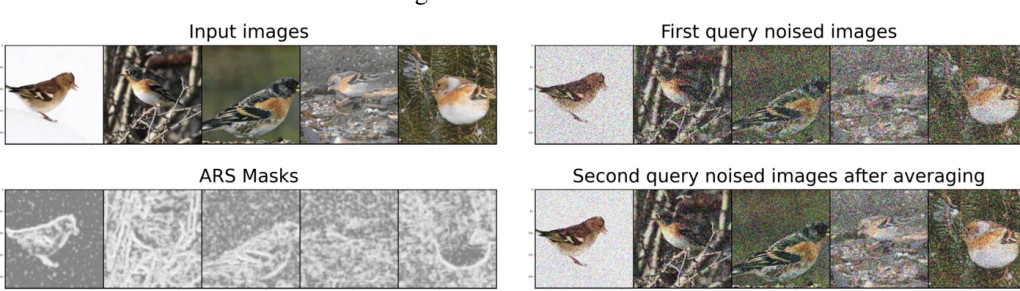

Figure 20: For $\sigma = 0.25$.

Figures 18 to 20 show how adaptive masking reduces the noise around areas that are important to classification for ImageNet. The images follow our architecture visualized Figure 1. The mask model is provided the first query noised images as input. The learned masks, presented in the bottom left, are sparse and concentrated around the area of interest (the bird, or any labelled object). The second query noised images after weighted average use the mask to clearly reduce the noise around the bird.

| Setting/Approach | Cohen et al. | ARS (Pretrain) | ARS (End-To-End) |
|---|---|---|---|
| ImageNet, $\sigma = 0.25$ | 68.5 (0.1) | 69.5 (0.0) | **70.1 (0.2)** |
| ImageNet, $\sigma = 0.5$ | 60.7 (0.5) | 63.2 (0.1) | **64.2 (0.2)** |
| ImageNet, $\sigma = 1.0$ | 47.9 (0.1) | 52.1 (0.0) | **53.8 (0.3)** |

Table 9: **Test accuracy without certification on ImageNet**.

An interesting observation is that on ImageNet, ARS yields larger improvements to the test accuracy without certification. We follow the certification procedure of Cohen et al. (2019), which first determines the predicted class, and then certifies it if and only if the probability of this predicted class is such that $p_+ \geq 0.5$ (that is, it groups all other classes into one non-predicted class). If this is not the case, the prediction will count as not certified at $r = 0$, even if the predicted class is still correct. We keep this procedure for consistency with prior work, but on a task with a large number of classes like in ImageNet, the accuracy at $r = 0$ can be much lower as the non-certified accuracy.

We noticed that ARS significantly improves this non certified accuracy, while not improving the accuracy at $r = 0$ as much (see Section 4.3). In effect ARS leads to more correct predictions, but

this (correct) predicted class has $\underline{p_+} < 0.5$, so the accuracy at $r = 0$ does not increase. Table 9 shows this effect by comparing the test accuracy (no certification at all) across all methods. Under ARS, the test accuracy increases from $68.5\%$ to $70.1\%$ when $\sigma = 0.25$, from $60.7\%$ to $64.2\%$ when $\sigma = 0.5$, and from $47.9\%$ to $53.8\%$ for $\sigma = 1.0$. In summary, ARS achieves best test accuracy (without certification) for all the noise levels. This suggests that ARS helps more than shown by the default certification approach, and that a finer analysis that accounts for the probability of all classes could yield further improvements in certified accuracy.

