# OpenReview forum: "Adaptive Randomized Smoothing: Certified Adversarial Robustness for Multi-Step Defences"
_NeurIPS.cc/2024/Conference — NeurIPS 2024 spotlight_

### Official Review · Reviewer_FN7A · 2024-06-13

**Soundness:** 4
**Presentation:** 3
**Contribution:** 4
**Rating:** 7
**Confidence:** 3

**Summary:**

The paper introduces Adaptive Randomized Smoothing (ARS), an innovative theoretical extension of Randomized Smoothing (RS) based on f-Differential Privacy (f-DP) to enhance the robustness certification of machine learning models. Empirical results demonstrate that ARS surpasses both standard RS and its variants in robustness certification metrics.

The authors present a solid theoretical framework and conduct thorough experimental evaluations. Despite ARS inheriting certain inherent constraints of RS, the method represents a significant advancement, offering a fresh approach to augment RS through budget decomposition.

**Strengths:**

1. The paper is well-written and easy to follow.
2. The theoretical underpinnings are solid: The integration of RS with f-DP and the subsequent decomposition leveraging f-DP properties is a creative and compelling strategy to refine RS.
3. The implementation of a masking mechanism is judiciously chosen to tackle the dimensionality challenges highlighted by the theoretical discourse.

**Weaknesses:**

1. As an enhancement of RS, ARS is subject to the same fundamental limitations, notably the requirement for a substantial sample size to attain satisfactory robustness certification. The authors acknowledge this constraint within the limitations section of the paper.

**Questions:**

1. Does the theoretical framework extend to alternative norm types, such as L1 or L2 norms?

**Limitations:**

See weaknesses and questions.

---

> ### Author Rebuttal · Authors · 2024-08-07
>
> Thank you for the thoughtful review. We are happy to answer the question about alternative norms.
>
> > Does the theoretical framework extend to alternative norm types, such as L1 or L2 norms?
>
> While the ARS theory (f-DP based RS and composition) applies to other norms such as L1 or L2, our specific instantiation using masking applies only to 𝐿∞ robustness:
>
> For L2: masking under the Gaussian mechanism implies no gains for L2 robustness directly. In particular, the reduction in the noise variance in the second step (Eq 6) is valid only against 𝐿∞ adversaries, and we would not be able to reduce noise this way for an L2 adversary (this is discussed in remark 3, ll. 177-181). Intuitively, this is because in the worst case, an L2 attack could be fully applied to pixels with a mask weight of 1, and so masking does not trigger noise reductions and our approach reduces to standard RS with two steps averaging the noise. This is a core reason why we do not present L2 results.
> For L1: bounding L1 with L2 (as we do between 𝐿∞ and L2 in the paper) does not involve the input dimension d. As a result, masking also doesn’t enable noise reduction by dimensionality reduction.
>
> That being said, it is plausible that by using multi-step composition, one could design architectures that improve robustness under other norms such as L1, L2, or L0. A potential avenue of research for L2 (that may apply to other norms as well) is to leverage the f-DP analysis of the subsampled Gaussian mechanism: https://arxiv.org/abs/1911.11607. This however would require significant design and analysis work, which is not yet explored, and in this submission we leave it for future work.

---

> > ### Comment · Reviewer_FN7A · 2024-08-09
> >
> > I greatly appreciate your discussion regarding the potential for extension to other norms. I think it would be most beneficial to include this revised discussion in the next version of your manuscript. I will keep my score for recommendation.

---

### Official Review · Reviewer_Zvfp · 2024-06-19

**Soundness:** 2
**Presentation:** 3
**Contribution:** 4
**Rating:** 6
**Confidence:** 4

**Summary:**

This work presents the first sound composition of randomized smoothing. Based on the novel theoretical results, this work presents the first sound way to compose a mask generator with the Gaussian sampling required for randomized smoothing, thereby reducing the effective dimension and improves the certification of randomized smoothing. This work opens up the possibility of more complex compositions for randomized smoothing, which is attempted before without soundness and great success. To this end, this work serves a good milestone in the field of randomized smoothing.

**Strengths:**

This paper presents a great theoretical contribution to randomized smoothing, namely the composition rule. I did not find evident mistakes in the proof, but others might spot issues. If the theorems presented are correct, I recommend immediate acceptance, regardless of the experimental inadequacy that I will elaborate in the weaknesses section.

The experimental section presents three different benchmarks, characterizing different aspects. The results show the advantage of the proposed method. The $L_\infty$ robustness of randomized smoothing is improved and depend less on the data dimension.

**Weaknesses:**

The main inadequacy I spot is its experimental design, specifically Section 4.3. While Section 4.1 and 4.2 are interesting and validates some of the claims made, Section 4.3 presents the main benchmark in the field of randomized smoothing. However, the authors only present the results on ImageNet with $\sigma=0.5$, while the main practice in the feild is to report results on \{cifar-10, ImageNet\} times \{$\sigma=0.25, 0.5, 1.0$\}. I request the authors to present the full benchmark for comparison.

In addition, to compare with Cohen et al., the masking trick could also be applied to certiify $L_2$ robustness. It is not clear why the authors only report $L_\infty$ robustness in the paper.

**Questions:**

See weaknesses.

**Limitations:**

The authors adequately addressed the limitations.

---

> ### Author Rebuttal · Authors · 2024-08-07
>
> Thank you for the thoughtful review. We address the two main requests listed under weaknesses:
>
> >  the main practice in the field is to report results on {cifar-10, ImageNet} times {𝜎=0.25,0.5,1.0}
> > “I request the authors to present the full benchmark for comparison.”
>
> Thank you for the suggestion. In the rebuttal PDF we report results for CIFAR-10 (Figures 10 & 11) and ImageNet (Figure 12) at these noise levels. We will update all the CIFAR-10 and ImageNet results in the paper to these standard noise levels with evaluation across three seeds. Due to time and computational constraints during the rebuttal phase we have run a single seed for ImageNet experiments.
>
> The CIFAR-10 results include include standard CIFAR-10 (without backgrounds, k=32) and larger background images (k=64). As expected, our approach only yields modest gains on CIFAR-10 without backgrounds, as the mask cannot provide much dimension reduction (although we still see that the masks select relevant parts of the image). With large backgrounds, the improvements are larger as shown in Figure 11 (b). These rebuttal results are an improvement after finding and fixing a pre-processing inconsistency that led to suboptimal masks (this issue only affected ARS, on CIFAR-10, and no other results).
>
> > “In addition, to compare with Cohen et al., the masking trick could also be applied to certify 𝐿2 robustness. It is not clear why the authors only report 𝐿∞ robustness in the paper.”
>
> While the ARS theory (f-DP based RS and composition) applies to L2, our specific instantiation using masking applies only to 𝐿∞ robustness, in the sense that it implies no gains for L2 robustness. In particular, the reduction of the noise variance in the second step (Eq 6) is valid only against 𝐿∞ adversaries, and we would not be able to reduce noise this way for an L2 adversary (this is discussed in remark 3, ll. 177-181). Intuitively, this is because in the worst case, an L2 attack could be fully applied to pixels with a mask weight of 1, and so masking does not trigger noise reductions and our approach reduces to standard RS with two steps averaging the noise. This is a core reason why we do not present L2 results.
>
> That being said, it is plausible that using multi-step composition, one could design architectures that improve L2 robustness (a potential avenue of research for that is to leverage the f-DP analysis of the subsampled Gaussian mechanism: https://arxiv.org/abs/1911.11607). This however would require significant design and analysis work, which is not yet explored, and in this submission we leave it for future work.

---

> > ### Comment · Reviewer_Zvfp · 2024-08-08
> >
> > Thanks for the detailed rebuttal from authors. This generally clears my previous concerns. I maintain my previous score.

---

### Official Review · Reviewer_7UWe · 2024-07-12

**Soundness:** 4
**Presentation:** 3
**Contribution:** 3
**Rating:** 8
**Confidence:** 4

**Summary:**

This paper proposes a framework to derive robustness certificates for smoothed classifiers based on f-differential privacy (f-DP). The framework achieves the same tight certificate for Gaussian smoothed classifiers as Cohen et al. (2019), while enabling analysis of adaptive multi-step smoothing mechanisms using the adaptive composition theorem for f-DP. To demonstrate the framework, the authors design a two-step smoothing mechanism that effectively performs adaptive (input-dependent) dimension reduction to achieve robustness against $\ell_\infty$-bounded perturbations. Experiments on three datasets demonstrate substantial improvement over baselines in some cases.

**Strengths:**

**Originality and significance:**
This paper reconnects randomized smoothing and differential privacy, showing that f-DP can achieve competitive robustness certificates. By identifying this connection, the paper could seed further innovations in randomized smoothing, where the design of smoothing mechanisms has been limited due to challenges in obtaining tractable robustness certificates. This has been a problem in prior work, where the analysis of smoothing mechanisms using input-dependent additive noise (e.g., Hong et al., 2022) were later shown to be unsound.

**Quality:**
The technical foundation is strong, which is important for work that aims to achieve provable robustness. The experiments are solid, covering a range of settings, including those where the adaptive masking approach does not achieve a significant performance advantage (ImageNet). In some cases the improvement in certified accuracy/standard accuracy over baselines is quite significant (of order 10 percentage points).

**Clarity:**
The paper was a pleasure to read.

**Weaknesses:**

The proposed adaptive masking smoothing mechanism does not achieve significant improvements in certified robustness over baselines in many cases, especially for CIFAR-10 (Fig 2) and ImageNet (Fig 6). There is a more pronounced improvement for CelebA, however I wonder if this is related to the fact that the masking model is trained under supervision or whether the masking smoothing mechanism is simply a better fit for CelebA? This could perhaps be investigated by repeating the experiment without using the mouth location metadata.

Given the (at times) marginal improvement of the adaptive masking smoothing mechanism, it would be interesting to understand how it compares along other dimensions, like computational efficiency and model size. It’s noted (line 324) that the mechanism is less efficient and uses a larger model, however it would be good to quantify to better understand the trade-offs.

Theorem 2.3 applies for a composition of randomized Gaussian mechanisms, which may be limiting given the range of mechanisms studied in the DP literature. I expect the theorem could be adapted for other mechanisms, but I wonder whether this would result in a tractable radius?

**Questions:**

- To better understand the generality of the approach: can a certified radius be derived for a composition of heterogeneous f-DP mechanisms (not necessarily all Gaussian)?

- Is there an intuitive explanation for the larger gap in certified accuracy between ARS and Cohen et al. for CelebA versus ImageNet?

- Are results available for a non-uniform split of the noise budget (line 223)?

**Limitations:**

Yes, limitations are addressed adequately in Section 5.

---

> ### Author Rebuttal · Authors · 2024-08-07
>
> Thank you for the thoughtful review. We start by answering the main questions in the review and then address further comments.
>
> > Q1. “[...] can a certified radius be derived for a composition of heterogeneous f-DP mechanisms [...]?”
>
> This is a great question! As noted in the review, the theory directly applies. However, there are two key elements to compute the bounds that are not necessarily trivial to get for general mechanisms: (1) the f-DP curve of a mechanism at any attack radius value r (to compute the sup line 471); (2) the composition of such curves across mechanisms (this is really easy for Gaussian mechanisms, but more challenging for other mechanisms even without heterogeneity).
>
> Fortunately, the DP community already has several tools that one could leverage to explore this design space. Notable examples that we believe are promising include:
> Using results from deep learning training with f-DP that study efficient composition of subsampled Gaussian mechanisms (https://arxiv.org/abs/1911.11607). This would help support steps that first subsample pixels and then apply the Gaussian mechanism (though some work is needed on the subsampling front).
> Using numerical tools for f-DP curves and composition (e.g., https://arxiv.org/abs/2106.08567, https://arxiv.org/abs/2106.02848) to support more complex mechanisms and heterogeneous composition.
>
> > Q2. “Is there an intuitive explanation for the larger gap in certified accuracy between ARS and Cohen et al. for CelebA versus ImageNet?”
>
> We would first like to address one possible reason suggested in the review:
>
> > “There is a more pronounced improvement for CelebA, however I wonder if this is related to the fact that the masking model is trained under supervision or whether the masking smoothing mechanism is simply a better fit for CelebA?”
>
> Thank you for the great suggestion. We re-ran our CelebA experiments, but without supervision for the mask, and it turns out that the supervision was not needed. The results are essentially unchanged (if anything, the variance is lower due to better masks) and visually the masks are a bit better (likely because our supervision was a coarse, approximate bounding box around the mouth, while masks without supervision are more precise). Please see Fig. 8 and 9  in the rebuttal PDF attached to the global response.
>
> As to why the results on CelebA are much better than on other tasks, we believe that it is because the CelebA task is very well suited to gains from dimension reduction, which our approach provides. Indeed, our task on CelebA has a fairly high input resolution for certification (160x160), while the relevant part for the prediction (the mouth) is often as small as 15x15. We can see in the mask images in the attachment that the mask model accurately selects the relevant mouth area, and the final image is much clearer at that position due to noise reduction from the lower dimension.
>
> We investigated why we did not see similar improvements on CIFAR-10 with large $k$. We found a pre-processing inconsistency between our steps that led to suboptimal masks (this only affected CIFAR-10 experiments for ARS and no other results). Fixing this led to crisper masks, and improved performance (see Figures 11a for results with $k=64$ that lead to gains similar as those seen for CelebA). On ImageNet, we do not expect such gains as the images are more complex, and relevant parts of the image are fairly large compared to the total dimensions.
>
> > Q3. “Are results available for a non-uniform split of the noise budget (line 223)?”
>
> We did not observe much gains from early experiments learning the split or using non-uniform splits, so we do not pursue this for this paper and do not have extensive results. However, it is plausible that more extensive tuning of this parameter can yield improvements, and it would be interesting to explore this avenue in more depth.
>
> > “It’s noted (line 324) that the mechanism is less efficient and uses a larger model, however it would be good to quantify to better understand the trade-offs.”
>
> We provide some numbers on ll. 271-273 of the submission (the time overhead for certification compared to Cohen et. al. is about 2x). We will add more details on mask model size and impact on training time and resources in that paragraph and Appendix D.

---

> > ### Comment · Reviewer_7UWe · 2024-08-08
> >
> > Thanks for the detailed responses to my questions. I have decided to raise my score for soundness and the overall rating. It will be interesting to see further exploration of this framework, particularly as advancements are made in f-DP.

---

### Official Review · Reviewer_NkYs · 2024-07-13

**Soundness:** 3
**Presentation:** 3
**Contribution:** 3
**Rating:** 7
**Confidence:** 4

**Summary:**

The paper uses tools from differential privacy (f-DP relating privacy to hypothesis testing, and also compositional rule of DP) to design a differentially private  2-step mechanism. The first step is interpreted here as creating a mask, and the second step is the "standard randomized smoothing" on the masked input, so it potentially greatly reduces the dimensionality.

**Strengths:**

The main result is Proposition 2.4, where it is shown that we can use two step randomized smoothing, where in the first step we effectively obtain a mask to mask out some of the input pixels, and by this effectivelly escaping the curse of dimensionality we have in randomized smoothing and the we use the standard smoothing on the masked input. The soundness is guaranteed by tools from differential privacy, framing the adversarial problem in the privacy language and use the standard (i suppose) tools to perform a private compositional test. The neighbouring databases (in the privacy sense) are formulated as $\ell_\infty$ distance in the image space, which exactly matches the setting of adversarial robustness (this was considered in the very first RS paper already that also took the DP approach to randomized smoothing).

The originality comes from bringing standard tools from a different field and demonstrates that they are effective here. Quality is high, the results and ideas are fresh and new in the field and improve on SotA. Clarity is somewhat good but not perfect, the authors seem to come from DP community (I'm from robustness) so It requires some effort and reading other papers for me to follow. Significance is on the higher end of the randomized smoothing papers in the recent years, the results might seem incremental on benchmarks, but the actual techniques are new so it will likely attract attention. The downside is not convincing empirical evaluation (is limited and in the standard settings the improvements are very mild, the main benefits are demonstrated on CELEBA benchmark), overall the empirical evaluation does not convince me that this approach outperforms SotA by a non-trivial margin. This being said, the work (mainly the tools used) is clearly interesting enough to be accepted.




very minor:
* f-DP is not really introduced I think, the $f$ appears in prop 2.1 without any explanation, right?
* unify Dong 2019 and Dong 2022 citations. I think they are the same?
* " adaptivity over multi-step computation comes with no increase in certified radius" - no decrease?

**Weaknesses:**

^

**Questions:**

-

**Limitations:**

-

---

> ### Author Rebuttal · Authors · 2024-08-07
>
> Thank you for the thoughtful review. We will add a background section on f-DP in the appendix and forward reference it before Prop 2.1 to provide more context. We will unify the citations to prefer the published edition (2022) over the arXiv edition (2019). Thank you for catching our increase/decrease typo! It is indeed no decrease.
>
> > [...] in the standard settings the improvements are very mild, the main benefits are demonstrated on CELEBA benchmark) [...]
> > This being said, the work (mainly the tools used) is clearly interesting enough to be accepted
>
> In the attached pdf, we provide further results for CIFAR-10 with no background and large background (to emphasize the importance of dimensionality). We investigated the degree of improvement on CIFAR-10, and found a pre-processing inconsistency between steps that led to suboptimal masks (this only affected CIFAR-10 experiments for ARS and no other results). Fixing this led to crisper masks, and improved performance at large $k$ (Figures 11b), where the results are now similar to those of CelebA. On ImageNet, we do not expect such gains as the relevant parts of the image are fairly large compared to the total dimensions. Please note the noise levels in these experiments follow the suggestion of Zvfp, and we will update all the paper results to these standard noise levels with evaluation across three seeds.

---

> > ### Comment · Reviewer_NkYs · 2024-08-07
> >
> > Thanks for the rebuttal.
> >
> > I think the experiment in 11 b is maybe a bit too artificial to convince me about the proposed method, but I am already convinced anyway :-) I keep my rating.

---

### Author Rebuttal · Authors · 2024-08-07

We thank all of the reviewers for their constructive reviews. We provide answers to questions in our review-wise responses, and we have completed experiments following the suggestions and requests that we include in the pdf attached to this global response.

Specifically:

- We provide CelebA experiments without mask supervision (reviewer 7UWe), and the results are as good (if not better, as the masks are sharper) than with our coarse mask supervision. See Figures 8 & 9 (the first two figures in the rebuttal pdf).
- For CIFAR-10, we investigated the degree of improvement, as highlighted by reviewers NkYs and 7UWe, and found a pre-processing inconsistency that led to suboptimal masks (this only affected ARS, on CIFAR-10 experiments, and no other results). Fixing this improves results both quantitatively and qualitatively. For $\sigma = 0.12, 0.5, 1.5$ and $k=48$ (bottom section in Table 1, last column), ARS standard certified accuracy (at $r=0$) improves to 83.66% from 82%, 66.6% from 64.8% and 35.5% from 34% with similar standard deviations as before. We also report results (masks and certified accuracy) for no background (k=32) and large background (k=64) to emphasize the importance of higher dimensionality. As expected, ARS provides real but modest gains in terms of certified accuracy on standard CIFAR-10 images (Figure 11a): the masks select relevant pixels, but dimensionality gains are small (Figure 10a). For $k=64$, the gains are much larger (Figures 11b), as the masks successfully ignore the high dimensional distractor background (Figure 10b).
- We provide results at the common noise levels of sigma = 0.25, 0.5, 1.0 (reviewer Zvfp) on CIFAR-10 (Figure 11), including without background (Figure 11a), and ImageNet (Figure 12). We will update all the paper results to these standard noise levels with evaluation across three seeds. Due to time and computational constraints during the rebuttal phase, we have run a single seed for ImageNet experiments.

---

### Decision · Program_Chairs · 2024-09-25

**Decision:**

Accept (spotlight)

**Comment:**

This paper introduces a two-step method for providing robustness certificates of smoothed classifiers using tools from differential privacy (f-DP) for multi-step composition of the guarantees. The proposed framework uses adaptive dimension reduction to improve robustness guarantees.

Certified robustness (CR) as a field began from connections to DP, and this paper returns to those roots. Reviewers found the ideas and tools to be very interesting, and their combinations to be novel. The reviewers also highlighted that the work was likely to spark the interest of CR researchers, which could lead to a broad impact on the subfield. I recommend that the authors add their discussion of alternative norms to the paper so that readers are clear of the limitations and avenues for future work.